# TRI-AGENT DRIVING: LEARNING TO COORDINATE AGENTS VIA SCENARIO COMPLEXITY REPRESENTATION FOR EFFICIENT AUTONOMOUS DRIVING

## ABSTRACT

End-to-End (E2E) autonomous driving systems face a fundamental dilemma that fast traditional models offer low latency but struggle with complex and ambiguous scenarios, while Vision-Language Models based systems provide powerful contextual understanding at the cost of high computational overhead. Instead of pursuing a single faster or more powerful model, we present Tri-Agent Driving (TAD), a dynamic framework that learns to select the most appropriate agent on-the-fly based on scenario complexity, directly from raw multi-view camera inputs. The predicted scenario complexity serves as a routing signal to enable real-time activation of the optimal agent, balancing computational efficiency and reasoning depth on demand. TAD integrates three complementary agents: a Fast Agent optimized for low-complexity and routine scenarios, a Smart Agent for medium-complexity scenarios, and a Deep Thinking Agent enhanced with Chain-of-Thought (CoT) reasoning for high-complexity corner cases. The core of TAD lies in the trainable Agent Coordination module, which proactively predicts scenario complexity and triggers agent switching without human intervention. On a challenging hybrid test set spanning diverse traffic conditions, TAD achieves state-of-the-art trajectory prediction, while reducing average inference latency by 26% (4.2s vs. 5.7s) and GPU memory consumption by 30% (15.4 GB vs. 22 GB) compared to the strongest VLM-based model on a single 3090 GPU. This "fast when possible, deep when necessary" paradigm establishes a new standard for efficient, robust, and adaptive end-to-end autonomous driving.

## 1 INTRODUCTION

E2E autonomous driving systems in the real world operate under a severely long-tailed distribution of traffic scenarios, where most situations are routine and easy to handle, while a small fraction involves rare but safety-critical cases, such as semantic ambiguity (temporary changes in traffic rules), dynamic multi-agent interactions (e.g., sudden lane-cutting by vehicles) and extreme corner cases (e.g., low light conditions). These safety-critical scenarios, though infrequent, account for the majority of system failures and demand deep semantic reasoning.

This inherent long-tailed nature calls for a dynamic resource allocation mechanism. In routine scenarios, the system should operate with minimal computational overhead, while in complex or ambiguous cases, it must be able to invoke more powerful reasoning capabilities on demand.

However, existing E2E approaches fall short of this ideal. On one hand, traditional architectures (Chen et al., 2024b; Huang et al., 2022; Jiang et al., 2023a) offer low latency and resource consumption (Fig. 1 (a)), but suffer from poor generalization to unseen or semantically ambiguous situations due to their reliance on training data distribution. On the other hand, Vision-Language Models (VLMs) (Sima et al., 2023; Tian et al., 2025) excel at semantic understanding via cross-modal alignment (Fig. 1 (b)), yet their massive parameter counts make them impractical for deployment across all scenarios, particularly in routine driving conditions. To bridge this gap, hybrid architectures have emerged. The first type is the dual system framework (Fig. 1 (c)), where large models assist the traditional ones (Jiang et al., 2024; Tian et al., 2025). However, these systems adopt a static fusion strategy, which is fixed to taking the VLM's output as auxiliary information input to traditional mod-

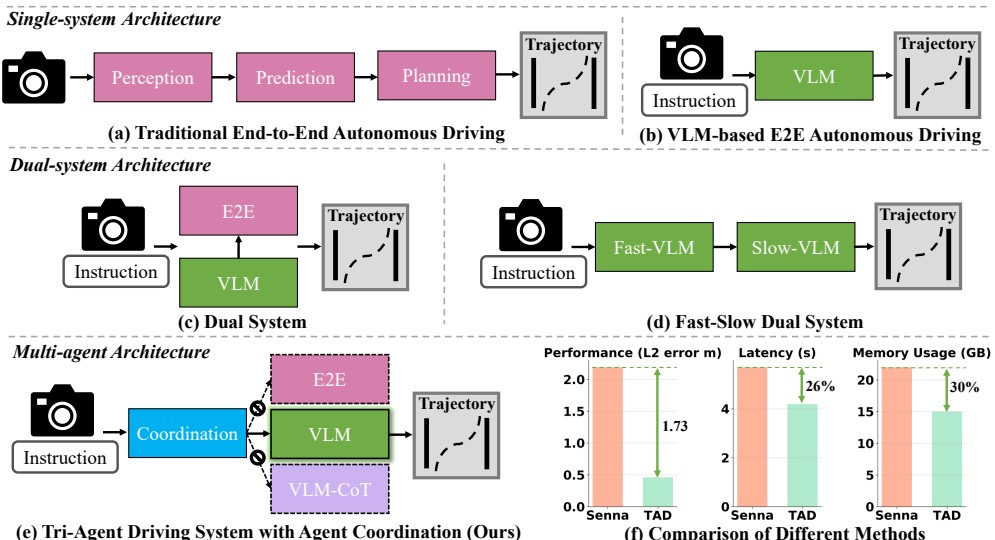

Figure 1: Comparison of existing autonomous driving frameworks and the advantages of the TAD system over existing methods in terms of performance, speed and GPU consumption.

ules, lacking the ability to adapt to dynamic changes in scenarios. The other type is fast-slow system (Fig. 1 (d)), which mimics human cognition by activating a "slow" reasoning module for corner cases (Zhang et al., 2025). However, existing solutions lack an automatic invocation mechanism, often relying on manual intervention or enforcing sequential execution, leading to computational redundancy and delayed responses.

To overcome these limitations, we propose Tri-Agent Driving (TAD), a novel learning framework that dynamically coordinates multiple autonomous agents by learning lightweight scenario complexity representation directly from raw multi-view images. This learned representation serves as a differentiable routing signal, enabling real-time and context-aware agent selection to optimally balance computational efficiency and reasoning depth.

The proposed TAD comprises three agents, specifically designed for different levels of scenario complexity. Fast Agent is optimized for low-complexity scenarios, prioritizing efficiency and real-time responses. Smart Agent is tailored for medium-complexity scenarios, striking a balance between computational efficiency and reasoning capability. Deep Thinking Agent is designed for high-complexity scenarios, using CoT (Wei et al., 2022) reasoning method to ensure safety and robustness. This tri-agent design is motivated by the observation that a large fraction of real-world driving scenarios falls into an intermediate regime, too complex for purely traditional pipelines, yet not demanding the full cost of deep VLM reasoning. The core of TAD lies in the trainable Agent Coordination module, which proactively predicts scenario complexity via our lightweight network ScpViT and triggers fully autonomous agent switching. ScpViT learns to route inputs to specialized "agents". This paradigm is particularly effective in autonomous driving, where computational budgets are constrained and scenario distributions are long-tailed. Unlike prior hybrid systems that statically fuse the VLM's output or rely on manual mode selection, TAD enables runtime-adaptive invocation of optimal agent according to scenario complexity level.

The key contributions of this work are summarized as follows:

- **First Unified Multi-Agent Framework.** TAD is the first framework to integrate three distinct capabilities, fast response (Fast Agent), strong semantics (Smart Agent) and deep reasoning (Deep Thinking Agent), within a unified architecture. This holistic design covers the full spectrum of driving scenarios and integrates the high efficiency of traditional architectures and the strong generalization of VLMs.

- **Learned Scenario Complexity-Aware Agent Coordination.** We introduce a lightweight ScpViT network that learns to predict scenario complexity from raw sensor inputs, serving as an autonomous routing mechanism for dynamic agent coordination. This enables seamless and real-time adaptation to evolving traffic conditions without human intervention.

- **State-of-the-Art Efficiency and Performance Trade-off.** On a challenging hybrid test set, TAD achieves a new SOTA in trajectory prediction, while reducing inference latency by **26%** (4.2s vs. 5.7s) and GPU memory consumption by **30%** (15.4 GB vs. 22 GB) compared to the strongest VLM-based method on a single 3090 GPU.

## 2 METHODOLOGY

### 2.1 PROBLEM DEFINITION

Given multi-view camera inputs $X = \{x_1, x_2, ..., x_N\}$, where each $x_i \in \mathbb{R}^{H \times W \times 3}$ denotes the RGB image from the $i-th$ view, the goal of E2E autonomous driving system is to predict a physically feasible and safe trajectory $\mathcal{T}$ over the next $T$ timesteps.

The key challenge lies in the long-tailed nature of real-world driving scenarios, where most situations are routine and can be handled efficiently, while a small fraction of safety-critical cases demand deep semantic reasoning. This leads to a fundamental tension between computational efficiency and generalization capability that existing E2E approaches fail to resolve. Instead of forcing one model to handle all scenarios, we propose a dynamic framework that learns to select the most appropriate agent on-the-fly based on scenario complexity, directly from raw multi-view camera inputs.

### 2.2 OVERVIEW OF TRI-AGENT DRIVING SYSTEM

**Overview.** As illustrated in Fig. 2, the TAD system consists of a scenario complexity–aware agent coordination module and three complementary autonomous driving agents. The coordination module first estimates the complexity of the current scenario from raw multi-view inputs and then dynamically activates the most suitable agent. This design avoids unnecessary invocation of heavy reasoning modules in routine cases while still ensuring robustness in safety-critical situations. In practice, a large fraction of real-world driving scenarios falls into an intermediate regime, too complex for purely traditional pipelines, yet not demanding the full cost of deep VLM reasoning. For example, ambiguous traffic signs or moderate pedestrian density cannot be reliably handled by a purely traditional pipeline but also do not justify the full overhead of CoT-based reasoning. The Smart Agent fills this gap by offering moderate semantic reasoning at much lower cost than the Deep Thinking Agent.

**Agent Coordination.** The coordination module, powered by our lightweight ScpViT network, predicts the scenario complexity level and routes the input accordingly. Unlike static fusion or manual triggers, this routing is fully autonomous and data-driven, ensuring adaptive resource allocation across heterogeneous scenarios.

**Complementary Tri-Agents.** The agents are specifically designed for different levels of scenario complexity. (1) Fast Agent is optimized for low-complexity scenarios, prioritizing efficiency and real-time responses. (2) Smart Agent is tailored for medium-complexity scenarios, striking a balance between computational efficiency and reasoning capability. (3) Deep Thinking Agent is designed for high-complexity scenarios, using CoT reasoning method to ensure safety and robustness.

### 2.3 SCENARIO COMPLEXITY-AWARE AGENT COORDINATION

#### 2.3.1 CURATION PIPELINE FOR DRIVING SCENARIO COMPLEXITY

**Scenario Complexity.** The complexity of driving scenarios is primarily influenced by three key factors. First, environmental conditions. Varying weather (e.g., rain, fog, snow) and illumination (e.g., daytime, nighttime) directly impact the robustness and reliability of perception systems. Second, the density and interaction patterns of dynamic and static entities, including vehicles, pedestrians, and obstacles, significantly increase the difficulty of decision making. Third, we highlight the distribution shift, a factor largely overlooked in prior work (Liu et al., 2024), which refers to semantic or structural shifts between training data and real-world test scenarios (e.g. unconventional obstacles or temporary traffic rule changes). This factor plays a decisive role in determining the generalizability and safety of autonomous driving systems under open world conditions. All the influence factors are listed in Table 1.

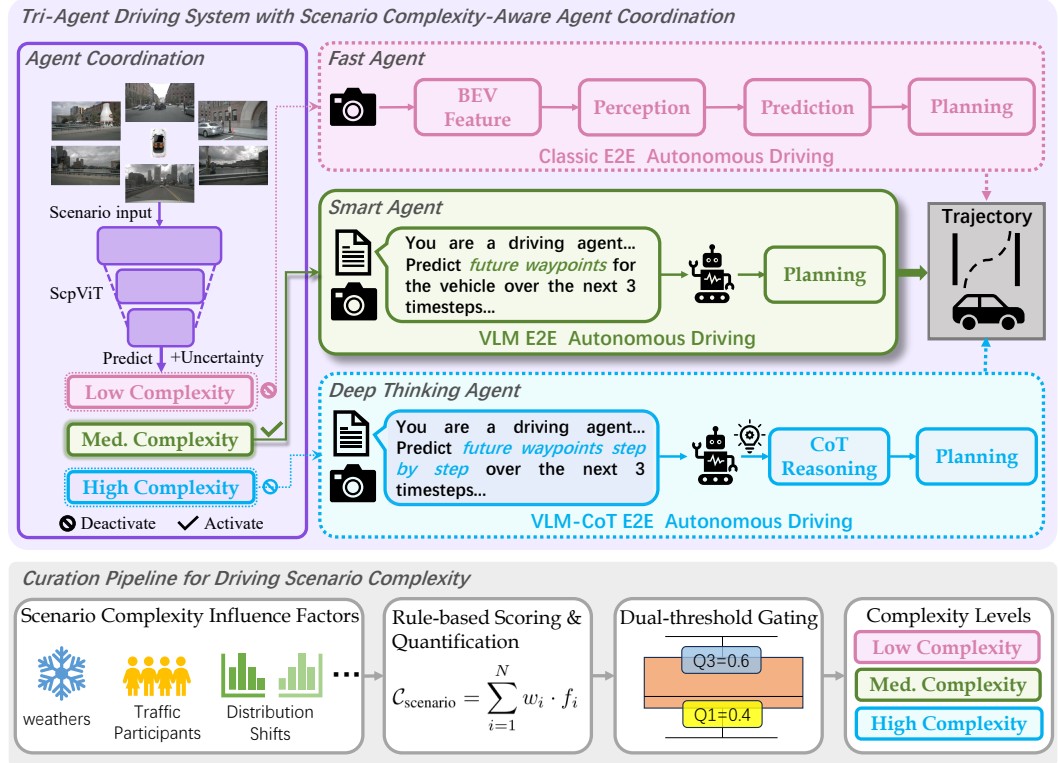

Figure 2: Overview of Tri-Agent Driving (TAD) and curation pipeline for driving scenario complexity. The first module is the Agent Coordination, which performs proactive scenario complexity prediction and uncertainty estimation, dynamically activating the most suitable autonomous driving agent according to the predicted complexity level. The remaining three modules are autonomous driving agents specifically designed for different levels of scenario complexity including Fast Agent, Smart Agent and Deep Thinking Agent. When the prediction uncertainty from ScpViT exceeds a predefined threshold, the system automatically falls back to the Deep Thinking Agent to provide more reliable decision-making in complex or ambiguous scenarios. The curation pipeline for driving scenario complexity includes scenario complexity scoring and a dual-threshold gating mechanism to help classify scenarios into different complexity levels.

The Scenario Complexity Score is computed as a weighted sum of multiple contributing factors:

$$\mathcal{C}_{\text{scenario}} = \sum_{i=1}^{N} w_i \cdot f_i, \tag{1}$$

where $\mathcal{C}_{\text{scenario}}$ denotes the complexity of the scene; $f_i \in [0, 1]$ represents the normalized score of the $i-th$ influence factor; $w_i$ is the corresponding weight assigned to factor $f_i$, satisfying $\sum_{i=1}^{N} w_i = 1$, and $N$ is the total number of factors considered.

**Dual-Threshold Gating Mechanism.** We further propose Dual-Threshold Gating Mechanism based on the quantile-based method to automatically classify different scenes into three complexity levels: *low*, *medium*, and *high*. This mechanism is to avoid manually defined absolute thresholds and instead dynamically determine the decision boundaries based on the distribution of the existing dataset, making it suitable for different tasks or datasets. Specifically, we use the 25th and 75th percentiles, denoted as $Q_1$ and $Q_3$, as the lower and upper thresholds:

$$T_{\text{low}} = Q_1 = F^{-1}(0.25), \quad T_{\text{high}} = Q_3 = F^{-1}(0.75), \tag{2}$$

where $F^{-1}(p)$ denotes the inverse of the cumulative distribution function (CDF) of the complexity score $c \in \mathbb{R}$. For a set of computed complexity scores $\mathcal{C} = \{c_1, c_2, \ldots, c_N\}$, the empirical quantiles can be obtained via sorting and interpolation:

$$Q_p = c_{(k)}, \quad k = \lceil p \cdot N \rceil. \tag{3}$$

Table 1: Factors influencing scenario complexity. The complexity index is the score assigned for different factors uniformly between 0 and 1.

| Scenario Complexity | | | |
|---|---|---|---|
| **Environment** | **Influence Factor** | **Value** | **Complexity Index** |
| Environment | Weather | Clear | 0.0 |
| | | Rainy | 0.5 |
| | | Fog | 1.0 |
| | | Snow | 1.0 |
| | Illumination | Normal | 0.0 |
| | | Low | 0.33 |
| | | Mid | 0.66 |
| | | High | 1.0 |
| Static/Dynamic Entities | Bounding Box Density | 0–5 | 0.33 |
| | | 6–10 | 0.66 |
| | | > 10 | 1.0 |
| | Types of Traffic Participants | 1-3 | 0.33 |
| | | 4-6 | 0.66 |
| | | > 6 | 1.0 |
| Corner Cases | Distribution Shift | Unconventional Obstacles | 1.0 |
| | | Traffic Rule Changes | 1.0 |
| | | Challenging Road Conditions | 1.0 |

Based on these thresholds, we define the complexity level of each sample as follows:

$$\text{Level}(c_i) = \begin{cases} \text{Low,} & \text{if } c_i < T_{\text{low}}; \\ \text{Medium,} & \text{if } T_{\text{low}} \leq c_i \leq T_{\text{high}}; \\ \text{High,} & \text{if } c_i > T_{\text{high}}. \end{cases} \quad (4)$$

As illustrated in the Fig.4 (b), we draw the Boxplot of all sample scores and overlay two critical threshold lines $T_{\text{low}}$ and $T_{\text{high}}$. The region under "Q1" is classified as *low complexity*, suitable for *Fast Agent* processing; the middle region represents *medium complexity*, requiring a balance between accuracy and efficiency, suitable for *Smart Agent* processing; and the region above "Q3" indicates *high complexity*, where full-capacity *Deep Thinking Agent* should be activated.

This mechanism leverages the statistical principle to build a parameter-free, interpretable, and distribution-adaptive complexity classification strategy which can be seamlessly integrated into dynamic inference frameworks to control computational resource allocation.

### 2.3.2 LIGHTWEIGHT NETWORK FOR SCENARIO COMPLEXITY PREDICTION

We propose ScpViT, a lightweight network designed for **S**cenario **C**omplexity **P**rediction within the Agent Coordination module. This module proactively forecasts the complexity of driving scenes to dynamically activate the appropriate autonomous driving agent. To meet real-time inference, we adopt MobileViTv2 (Mehta & Rastegari, 2022b), a highly efficient lightweight architecture as our base model. However, such compact networks often rely on operations like depthwise convolutions (Sandler et al., 2018) to save computation, which may discard fine-grained spatial details critical for accurate complexity estimation. To mitigate this, we integrate a parameter-free attention mechanism (Yang et al., 2021) that enhances feature representation without introducing additional parameters or computational overhead, effectively preserving crucial object-level cues. The improved architecture is illustrated in Fig. 3 (b).

The ScpViT network is trained in a supervised classification manner, where the ground-truth labels are the discretized scenario complexity levels (Low, Medium, High) computed via our curation pipeline (Section 2.3.1). Specifically, we treat complexity prediction as a 3-class classification task and optimize with cross-entropy loss.

To further enhance the system's robustness and safety, we incorporate an **Uncertainty-Aware Fallback Mechanism** (UAFM) when the prediction uncertainty from ScpViT exceeds a predefined threshold, the system automatically falls back to the Deep Thinking Agent, which offers stronger

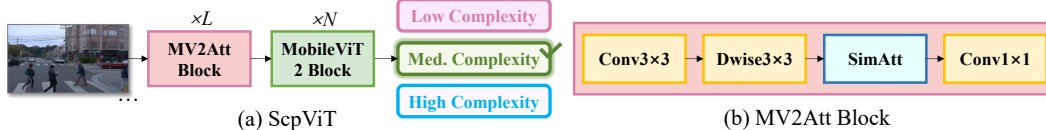

(a) ScpViT                                              (b) MV2Att Block

Figure 3: (a) The proposed lightweight network, ScpViT, for Scenario Complexity Prediction in Agent Coordination module. (b) The detailed structure of MV2Att Block, which is the refined version of Mobilenet V2 Block (Sandler et al., 2018).

generalization and reasoning capabilities, to provide more reliable decision-making in complex or ambiguous scenarios.

Given the output of the predicted class probability distribution $\mathbf{p} = (p_1, p_2, p_3)$ by ScpViT on three classes, the predictive uncertainty is quantified using the Shannon entropy:

$$\mathcal{H}(\mathbf{p}) = -\sum_{i=1}^{3} p_i \log p_i, \tag{5}$$

where higher entropy indicates greater uncertainty in the model's prediction.

To select the uncertainty threshold $\tau$ for our fallback mechanism, we evaluate a grid of candidate entropy and confidence thresholds. For each candidate pair, the validation samples are partitioned into high- and low-uncertainty groups, and the prediction error rates of both groups are compared. The optimal threshold is selected as the one that (1) the error-rate gap between high- and low-uncertainty samples and (2) the proportion of fallback-triggered cases (constrained to 10%–20%). This ensures the system falls back to the Deep Thinking Agent only when uncertainty strongly correlates with prediction failure.

## 2.4 COMPLEMENTARY TRI-AGENTS

The Complementary Tri-Agents integrate traditional autonomous driving systems and VLMs-based reasoning systems in parallel, activated dynamically by the Agent Coordination module according to scenario complexity. This design strategically leverages the core strengths of each component: traditional models has efficient, low-latency real-time responses in low-complexity scenarios, while VLMs unleash their powerful semantic reasoning and superior generalization capabilities in medium- and high- complexity environments. Through collaboration, the system achieves an optimal trade-off among efficiency, safety and adaptability.

**Fast Agent.** We adopt VAD (Jiang et al., 2023a) to ensure real-time response for routine driving scenarios, leveraging its advantages in real-time performance and structured scene representation.

**Smart Agent.** Traditional driving systems models often lack sufficient robustness when encountering novel or ambiguous situations. While, VLMs serve as a high-level semantic carrier that can express traffic rules, social commonsense, environmental intent and long-term dependencies, injecting "implicit knowledge" that is rarely captured by conventional labeled data. For example, proactively slowing down when seeing a "School" sign. This behavior relies not only on visual perception but also on understanding cross-modal semantic associations. Therefore, the deep integration of language with driving tasks not only expands the semantic boundaries of perception systems but also establishes a new paradigm for building intelligent driving systems that are interpretable and generalizable. In this paper, we use the VLM trained in the work (Chi et al., 2025) for medium-complexity situations, which supports scene understanding, prediction, and trajectory planning.

**Deep Thinking Agent.** The *Deep Thinking Agent* is designed to handle extreme corner cases that demand sophisticated reasoning, causal understanding and analysis, where both traditional models and standard VLMs may fail due to insufficient context modeling or lack of explicit logical inference. To enable reasoning in complex driving scenarios, we introduce CoT reasoning (Wei et al., 2022) and propose the VLM-CoT Agent, which performs step-by-step inference to ultimately generate safe and physically feasible trajectories. The reasoning process is formally decomposed into

three stages. (1) Perception and Intent Understanding. The agent first interprets the visual scene and infers high-level driving intent by answering: "What contextual cues are present, and what driving behavior is contextually appropriate?" (2) Dynamics and Feasibility Assessment. It then evaluates the safety of the inferred intent by reasoning: "Is the intended action dynamically feasible and collision-free under current traffic interactions and physical constraints?" (3) Execution and Trajectory Output. Finally, the agent translates the validated intent into a concrete motion plan by resolving: "How should the ego-vehicle execute this intent? Here is the optimized trajectory." The illustrative samples can be found in Fig. 5. This staged CoT framework endows the VLM-based agent with interpretable, safety-aware decision-making capabilities to process the complex driving scenarios.

In general, we summarize our approach in Algorithm 1.

---

**Algorithm 1** Tri-Agent Driving (TAD)

---

**Input:** Multi-view camera inputs $X = \{x_1, x_2, \ldots, x_N\}$, where $x_i \in \mathbb{R}^{H \times W \times 3}$.

**Parameters:** Fast Agent $\mathcal{A}_{\text{Fast}}$, Smart Agent based on VLM auxiliary system $\mathcal{A}_{\text{Smart}}$, Deep Thinking Agent capable of CoT reasoning $\mathcal{A}_{\text{Deep}}$, Scenario Complexity-Aware Agent Coordination $C_{\text{agent}}$, Uncertainty threshold $\tau$.

**Output:** Autonomous driving trajectory $\mathcal{T}$.

 1: Initialize inference status: `Inference` $\leftarrow$ `True`
 2: **while** `Inference` **do**
 3:    Step 1: Predict scenario complexity and confidence
 4:    $u$: predictive entropy
 5:    $\text{Level}_c(X), u \leftarrow C_{\text{agent}}(X)$
 6:    Step 2: Uncertainty-aware fallback
 7:    **if** $u > \tau$ **then**
 8:       $\mathcal{T} \leftarrow \mathcal{A}_{\text{Deep}}(X)$
 9:       **continue**
10:    **end if**
11:    Step 3: Complexity-based routing
12:    **if** $\text{Level}_c(X) = \text{High}$ **then**
13:       $\mathcal{T} \leftarrow \mathcal{A}_{\text{Deep}}(X)$
14:    **else if** $\text{Level}_c(X) = \text{Medium}$ **then**
15:       $\mathcal{T} \leftarrow \mathcal{A}_{\text{Smart}}(X)$
16:    **else**
17:       $\mathcal{T} \leftarrow \mathcal{A}_{\text{Fast}}(X)$
18:    **end if**
19: **end while**

---

## 3 EXPERIMENTS

### 3.1 DATASETS

To comprehensively evaluate the TAD system, we first construct a hybrid dataset composed of driving scenarios with varying levels of complexity. This dataset integrates the nuScenes dataset (Caesar et al., 2020), which provides diverse and representative driving scenarios; the nuScenes-C dataset (Xie et al., 2023), which specializes in challenging edge cases; and the Impromptu VLA dataset (Chi et al., 2025), which captures complex semantic and dynamic interactions. Scenario complexity scores are first computed using the rules in the Table 1 and Equation 1 and then the samples are categorized into low, medium and high complexity levels based on the dual-threshold mechanism (Section 2.3.1). For high-complexity scenes, we further annotate CoT reasoning annotations to fine-tune VLMs. The hybrid dataset statistics for the training and test splits are shown in Fig. 4 (a). Fig. 4 (b) illustrates the distribution of Scenario Complexity levels in the entire test set, categorized with the dual-threshold mechanism. Representative examples of scenarios across different complexity levels and the sample of CoT reasoning data are illustrated in Fig. 5. All the experiments will be implemented on this hybrid dataset.

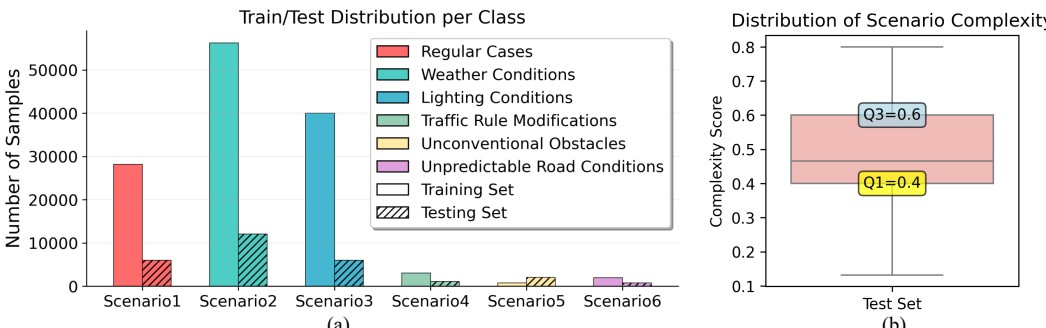

Figure 4: (a) The distribution of training and testing samples across different scenario categories in the hybrid dataset, which is composed of multiple sub-datasets. (b) Distribution of complexity scores with Dual-Threshold Gating. "Q1" and "Q3" denote the 25th and 75th percentiles to classify samples into low, medium and high complexity levels.

In addition, we also make closed-loop evaluation on CARLA dataset. CARLA Dev10 (Jia et al., 2025) is a recently introduced set of challenging evaluation routes. To enable efficient closed-loop validation, we augment it with an additional 28 sampled routes from CARLA, forming a compact test set, CARLA-Tiny.

### 3.2 EVALUATION METRICS

We conduct thorough assessments on both the Scenario Complexity Prediction task and the TAD autonomous driving system. First, for Scenario Complexity Prediction, we perform detailed evaluations across multiple dimensions, including classification accuracy across different complexity levels (low, medium and high) (**Acc.%**), and inference latency (**ms**). Furthermore, we conduct a comprehensive evaluation of the overall TAD system in terms of trajectory prediction accuracy (**L2 error**), inference latency (**ms**), and resource consumption (**MB** and **GB**), validating its effectiveness and scalability in real-world deployment scenarios.

For closed-loop evaluation, we adopt Route Completion (RC), and Driving Score (DS) (Shao et al., 2023a). The route completion refers to the percentage of the total route length that has been completed. It only takes into account the distance traveled along the predetermined route, where each segment of the predetermined route corresponds to a navigation instruction. If the agent deviates too far from the route, the agent is regarded as violating the instruction, and this episode is marked as a failure and terminated. The driving score is the product of the route completion ratio and the infraction score, describing both driving progress and safety.

### 3.3 EVALUATION OF SCENARIO COMPLEXITY-AWARE AGENT COORDINATION

We first evaluate the performance of the Scenario Complexity-Aware Agent Coordination module. At its core lies the Scenario Complexity Prediction task, a lightweight yet critical component that determines agent selection. To assess its efficiency and accuracy, we benchmark our proposed lightweight network, ScpViT, against other small models like MobileNet (Howard et al., 2017; 2019; Sandler et al., 2018) and MobileViT (Mehta & Rastegari, 2022a). The quantitative results are summarized in Table 2, demonstrating the superior performance of the proposed ScpViT.

### 3.4 EVALUATION OF TAD SYSTEM

**Open-Loop Evaluation.** To thoroughly evaluate our TAD system, we first evaluate the trajectory prediction performance of different agents within the TAD system across varying levels of scenario complexity. The results are summarized in Table 3. Experimental results show that both the Smart Agent and the Deep Thinking Agent perform well in low- and medium-complexity scenarios. In high-complexity settings, the Smart Agent[*] achieves moderate gains after finetuning on high-complexity data. In contrast, the Deep Thinking Agent further finetuned with our CoT strategy, demonstrates significantly greater performance improvements.

Table 2: Comparison of Models with Different Backbones for Scenario Complexity Prediction.

| Model Type | Feature Backbone | #Params | #Latency (ms) | Accuracy | | | |
|---|---|---|---|---|---|---|---|
| | | | | Low | Medium | High | Avg. |
| CNNs | MobileNetv1 | 2.6M | 11.3 | 72.15% | 87.79% | 70.05% | 82.82% |
| | MobileNetv2 | 2.6M | 10.2 | 68.13% | 99.79% | 75.41% | 90.85% |
| | MobileNetv3 | 2.5M | 11.3 | 72.42% | 98.86% | 58.08% | 89.41% |
| Transformers | ViT | 5.8M | 9.9 | 75.48% | 87.38% | 51.70% | 81.45% |
| | MobileViTv1 | 2.3M | 11.2 | 68.57% | 97.98% | 83.85 % | 90.50% |
| | MobileViTv2 | 5.6M | 12.7 | 70.35% | 99.05% | 84.39% | 91.67% |
| | ScpViT(Ours) | 5.6M | 12.7 | 70.32% | **99.60%** | **89.99%** | **92.59%** |

Table 3: Performance of our TAD system on test sets with varying levels of scenario complexity. The Fast Agent corresponds to the base version of VAD. For **TAD**, the Smart Agent is adapted from the model proposed in (Chi et al., 2025), trained on the nuScenes dataset (Caesar et al., 2020). For **TAD**$^*$, the Smart Agent$^*$ is sourced from (Chi et al., 2025), but trained on a combined dataset of both nuScenes (Caesar et al., 2020) and Impromptu (Chi et al., 2025). The Deep Thinking Agent is obtained by fine-tuning the Smart Agent$^*$ on our CoT annotated data.

| Agent | Traj. Pred. L2 Error (m) | | | | | | | | | | | |
|---|---|---|---|---|---|---|---|---|---|---|---|---|
| | Low Complexity | | | | Med. Complexity | | | | High Complexity | | | |
| | 1s | 2s | 3s | Avg. | 1s | 2s | 3s | Avg. | 1s | 2s | 3s | Avg. |
| **TAD** | | | | | | | | | | | | |
| Fast Agent | 0.41 | 0.70 | 1.05 | 0.72 | 0.52 | 0.88 | 1.32 | 0.91 | 3.46 | 5.72 | 7.94 | 5.71 |
| Smart Agent | 0.14 | 0.30 | 0.58 | 0.34 | 0.14 | 0.30 | 0.55 | 0.33 | 3.10 | 4.90 | 6.78 | 4.93 |
| Deep Thinking Agent | 0.17 | 0.35 | 0.64 | 0.39 | 0.12 | 0.28 | 0.53 | 0.31 | 0.68 | 1.25 | 1.96 | 1.30 |
| **TAD**$^*$ | | | | | | | | | | | | |
| Fast Agent | 0.41 | 0.70 | 1.05 | 0.72 | 0.52 | 0.88 | 1.32 | 0.91 | 3.46 | 5.72 | 7.94 | 5.71 |
| Smart Agent$^*$ | 0.13 | 0.27 | 0.52 | 0.30 | 0.13 | 0.28 | 0.53 | 0.31 | 1.28 | 2.12 | 3.11 | 2.17 |
| Deep Thinking Agent | 0.17 | 0.35 | 0.64 | 0.39 | 0.12 | 0.28 | 0.53 | 0.31 | **0.68** | **1.25** | **1.96** | **1.30** |

Subsequently, we conduct further comprehensive multidimensional comparison between the TAD system and various baselines, including traditional autonomous driving models, large language models adapted for autonomous driving and other dual-system approaches. The results are presented in Table 4. Experimental results show that, compared with single or dual systems, the proposed TAD exhibits significant advantages in both performance and speed. When the Agent Coordination with "looking ahead" capability achieves a 100% prediction accuracy, the system attains the lowest response latency and resource consumption. In the actual test scenarios where the prediction accuracy of agent coordination is 92.59%, the system still maintains a favorable balance between performance and speed. This also proves that complexity-aware routing mechanism is superior to using the best agent only. Furthermore, compared with existing dual-agent system, Senna, **TAD**$^*$(92.59%) performs better in all the key metrics. As can be seen, the L2 error of trajectory prediction is reduced by **1.73**, the inference latency is decreased by about **26%**, and the GPU memory consumption is saved by about **30%**. The proposed multi-agent system fully leverages the strengths of each subsystem and incorporates optimizations for complex scenarios, enhancing the system's adaptability under diverse conditions. In addition, we enhance TAD* with an entropy-based uncertainty estimation mechanism, when prediction uncertainty is high, the system falls back to the Deep Agent for safe inference. As shown in the table, this strategy reduces the average L2 error from 0.46 to 0.40 (a 13% improvement) with only a modest increase in latency (+0.6s) and memory usage (+1.4GB), effectively balancing safety and accuracy.

**Closed-Loop Evaluation.** We incorporate multiple baseline methods and conduct closed-loop experiments on the compact test set, CARLA-Tiny, evaluated with more comprehensive metrics. The results are presented in the Table 5. As shown in the table, TAD achieves an RC of 64.34, representing a 4.83 percentage point improvement over the strongest baseline model, Impromptu and its DS is 42.50, also higher (+0.67).

Table 4: Comparison of Trajectory Prediction for Different Methods. We report the average performance on the **hybrid test set** (including data with different scenario complexity levels). For Large Models, we have measured the average end-to-end latency on the test set. Impromptu-3B denotes the model is pretrained with nuScenes only. Impromptu-3B* denotes the model is pretrained with nuScenes (Caesar et al., 2020) and Impromptu (Chi et al., 2025) datasets. "Oracle, 100%" refers to the ideal situation where the proposed ScpViT perfectly classifies all samples' scenario complexity correctly.

| Method | Traj. Pred. L2 Error (m) ↓ | | | | Average E2E Latency (s) ↓ | Average Memory ↓ |
|---|---|---|---|---|---|---|
| | 1s | 2s | 3s | Avg. | | |
| *Autonomous Driving Models (Non-large Models)* | | | | | | |
| UniAD | 1.71 | 2.87 | 3.31 | 2.63 | 0.56 | 4228 MB |
| VAD | 1.19 | 1.99 | 2.83 | 2.00 | 0.22 | 3488 MB |
| *VLMs for Autonomous Driving Models* | | | | | | |
| Qwen2.5-VL-3B-Instruct | 3.09 | 5.39 | 7.10 | 5.19 | 5.6 | ≈ 19GB |
| OpenEMMA-3B | 1.30 | 2.10 | 2.98 | 2.14 | 5.2 | ≈ 19GB |
| Impromptu-3B | 0.43 | 0.75 | 1.16 | 0.78 | 5.4 | ≈ 19GB |
| Impromptu-3B* | 0.24 | 0.46 | 0.78 | 0.49 | 5.4 | ≈ 19GB |
| *Hybrid Systems for Autonomous Driving Models* | | | | | | |
| Senna | 1.37 | 2.17 | 3.03 | 2.19 | 5.7 | ≈ 22GB |
| **TAD**(Oracle, 100%) | 0.23 | 0.45 | 0.76 | 0.48 | 3.7 | ≈ 13.8GB |
| **TAD**(92.59%) | 0.24 | 0.47 | 0.79 | 0.50 | 4.2 | ≈ 15.4GB |
| **TAD**\*(Oracle, 100%) | 0.23 | 0.44 | 0.75 | 0.47 | 3.7 | ≈ 13.8GB |
| **TAD**\*(92.59%) | 0.22 | 0.43 | 0.73 | 0.46 | 4.2 | ≈ 15.4GB |
| **TAD**\*(92.59%) + **UAFM** | 0.18 | 0.37 | 0.65 | 0.40 | 4.8 | ≈ 16.8GB |

Table 5: Closed-loop evaluation of different methods on CARLA-Tiny.

| Method | Route Completion (RC) ↑ | Driving Score (DS) ↑ |
|---|---|---|
| AD-MLP (Zhai et al., 2023) | 0.00 | 13.05 |
| UniAD-Base (Hu et al., 2023) | 51.81 | 36.12 |
| VAD-Base (Jiang et al., 2023a) | 48.48 | 34.66 |
| Qwen3-2B (Yang et al., 2025) | 58.69 | 40.70 |
| Impromptu-3B (Chi et al., 2025) | 59.51 | 41.83 |
| TAD (Ours, 100%) | **64.34** | **42.50** |

# 4 RELATED WORK

The architectural design of perception and decision-making systems has long stood as a central pillar in autonomous driving research. Multiple architectural paradigms have emerged and these primarily fall into three categories: traditional modular pipelines, which decompose the driving task into discrete, interpretable stages; end-to-end systems powered by Vision-Language Models (VLMs), which map raw sensor inputs directly to trajectories; and hybrid architectures that strategically combine the strengths of both paradigms. **A comprehensive survey of these three perspectives is provided in the Appendix B.**

# 5 CONCLUSION

In this paper, we propose Tri-Agent Driving, an efficient and robust end-to-end autonomous driving system via scenario complexity-aware agent coordination. This system provides a feasible and effective solution for balancing efficiency and generalization ability in autonomous driving, and also offers new insights for building more intelligent and flexible autonomous driving architectures in the future. Currently, the scenario complexity predictor is trained offline. In future work, we plan to explore joint end-to-end training with the downstream agents, where the routing decision is optimized via reinforcement learning.

## ETHICS STATEMENT

This work complies with the ICLR Code of Ethics. No ethical concerns requiring specific disclosure are present.

## REPRODUCIBILITY STATEMENT

To ensure reproducibility, we provide: (1) implementation details in the Appendix C; (2) the training protocols follow the works (Mehta et al., 2022; Mehta & Rastegari, 2022a; Chi et al., 2025), of which the codes are all released; (3) the training and inference code upon paper acceptance.

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

## A  THE USE OF LARGE LANGUAGE MODELS (LLMS)

Firstly, Large Language Models were used as a writing assistance tool in this work. Their role was strictly limited to grammar checking, sentence polishing, and improving the fluency and clarity of the manuscript's language. The models did not contribute to research ideas, experimental design, and data analysis. All technical results, claims, and conclusions remain the sole responsibility of the authors.

Secondly, the CoT data used for fine-tuning the Deep Thinking Agent was generated with the assistance of a large language model. We provided the model with original scene annotations and a structured reasoning template to produce the step-by-step rationales. The model's role was strictly limited to generating these textual explanations based on existing data; it did not contribute to the design, or analysis of the research. All final CoT annotations were reviewed for consistency and plausibility.

## B  RELATED WORKS

The architectural design of perception and decision-making systems has long stood as a central pillar in autonomous driving research. Multiple architectural paradigms have emerged and these primarily fall into three categories: (1) traditional modular pipelines, which decompose the driving task into discrete, interpretable stages; (2) end-to-end systems powered by Vision-Language Models (VLMs), which map raw sensor inputs directly to trajectories; and (3) hybrid architectures that strategically combine the strengths of both paradigms. The following is a review of existing research from these three perspectives.

### B.1  MODULAR AUTONOMOUS DRIVING

The dominant paradigm in industrial autonomous driving systems remains the modular pipeline, which is typically decomposed into perception, prediction, and planning. This architecture offers strong real-time response and mature deployment. However, its inherently decoupled structure introduces critical limitations: information degrades as it passes between modules, expressive capacity is constrained by hand-designed interfaces, and errors tend to cascade downstream, making it difficult to meet the demands of efficient decision-making in highly interactive and complex dynamic scenarios.

To address the semantic inconsistency and response delay caused by module decoupling, recent research has shifted toward end-to-end (E2E) learning frameworks that jointly model and optimize the perception-prediction-planning pipeline. End-to-end autonomous driving (Hu et al., 2022; Shao et al., 2023b; Chen et al., 2024b; Li et al., 2024; 2025; Liao et al., 2024; Hu et al., 2023; Jiang et al., 2023a) has emerged as a hotspot due to its ability to directly learn driving strategies from raw sensor input. By unifying traditionally separate modules into a single differentiable system, E2E architectures minimize error accumulation, enhance safety through holistic optimization, and offer a scalable, data-driven alternative to rigid, rule-based planners. Importantly, such systems maintain real-time inference capabilities, facilitating practical deployment. Notably, VAD (Jiang et al., 2023a) pioneers a vectorized scene representation that fuses perception and HD map cues into a structured, learnable vector space, enabling joint end-to-end trajectory prediction and motion planning. This approach significantly boosts response efficiency and lays a promising foundation for integrated, structured modeling in autonomous systems. Nevertheless, despite these advances, current E2E systems often rely on predefined feature representations and static architectures, which can hinder generalization when confronted with novel or highly complex traffic scenarios.

## B.2 Vision-Language Models for End-to-End Autonomous Driving

With the rise of Vision-Language Models (VLMs), more and more studies have attempted to apply VLMs to autonomous driving tasks. By jointly embedding visual observations and linguistic reasoning into a unified semantic space, Large Language Models (LLMs) and Vision-Language Models (VLMs) are driving the autonomous driving and robotic systems from passive perception to active understanding, emerging as a new way toward building interpretable and generalizable intelligent agents. The integration of natural language priors with driving tasks unlocks a powerful new avenue for enhancing autonomous systems: it enriches perceptual reasoning through semantic grounding, improves decision transparency via language-based explanations, and boosts adaptability in complex, open-world, or previously unseen scenarios. As such, VLMs are emerging not merely as tools for perception, but as cognitive scaffolds that enable higher-level scene comprehension and intent-aware decision-making, paving the way for the next generation of robust, human-aligned autonomous agents.

Studies (Hwang et al., 2024) have explored end-to-end autonomous driving systems based on pre-trained models such as Qwen or Gemini to directly output the trajectories. Qian et al. (2025) introduce AgentThink, a pioneering unified framework that, for the first time, integrates Chain-of-Thought (CoT) reasoning with dynamic, agent-style tool invocation for autonomous driving tasks. AutoVLA (Zhou et al., 2025) is a novel Vision-Language-Action (VLA) model that unifies semantic reasoning and physically feasible trajectory planning within a single autoregressive framework. It first leverages VLMs to generate high-level action commands, which then serve as semantic priors to guide the prediction of trajectories. By retaining the vision-language model as its architectural backbone, AutoVLA enables optimization between action semantics and trajectory planning, significantly enhancing both the interpretability and physical plausibility of driving decisions. This model supports two reasoning modes: fast thinking (direct trajectory generation) and slow thinking (augmented with chain-of-thought reasoning for complex scenarios). However, the current version lacks an automatic mechanism to dynamically switch between these modes. Mode selection must be performed by drivers.

Besides, these methods face another two common challenges. First, the inference process requires large computation and causes a high response delay, making it difficult to meet the real-time requirements of autonomous driving. Second, its "black-box" nature leads to a lack of interpretability in the system, which is not conducive to problem diagnosis.

## B.3 Hybrid Architectures

Several studies have explored hybrid architectures that integrate traditional modular pipelines with end-to-end Vision-Language Models (VLMs) to balance interpretability, efficiency, and semantic richness. DriveVLM (Tian et al., 2025) and Senna (Jiang et al., 2024) augment conventional planning by incorporating VLM-generated low-frequency trajectories or high-level semantic guidance. However, these approaches rely on static fusion strategies, treating VLM outputs as fixed auxiliary inputs to downstream modules, which limits their capacity to adapt dynamically to evolving scene complexity. Moreover, they often fail to explicitly differentiate between routine driving conditions and rare corner cases. This leads to suboptimal resource allocation: VLMs are underutilized in critical, ambiguous scenarios where their reasoning capabilities are most valuable, while being redundantly invoked in simple contexts, compromising efficiency. Besides, the inherently high inference latency of VLMs introduces non-negligible system-level delays compared to purely modular architectures, potentially undermining real-time responsiveness.

The proposed Tri-Agent Driving (TAD) framework is different from prior approaches in several key aspects. First, TAD seamlessly integrates Vision-Language Models (VLMs) with traditional modular pipelines, enabling runtime-adaptive invocation of VLM-based reasoning: activating high-capacity semantic understanding only when scenario complexity demands it. At its core, TAD introduces a scene complexity–aware agent coordination mechanism that dynamically selects the most appropriate agent to handle different driving scenarios. This ensures optimal trade-offs between efficiency and performance. Critically, mode switching is fully autonomous. The system triggers agent transitions in real time based on contextual cues, requiring no human intervention.

## C    IMPLEMENTATION DETAILS

For the Scenario Complexity Prediction task, we follow the same training protocol in the works (Mehta & Rastegari, 2022a) to ensure a fair comparison. The training and testing are conducted on 3090 GPU. We train our ScpViT for 10 epochs with an effective batch size of 1024 images (128 images per GPU × 8) using AdamW. We linearly increase the learning rate from $10^{-6}$ to 0.002 for the first 20k iterations. After that, the learning rate is decayed using a cosine annealing policy. We implement our models using CVNets (Mehta et al., 2022), and use their scripts for data processing, training, and evaluation.

The TAD system primarily integrates three subsystems: VAD (Jiang et al., 2023a), Impromptu (Chi et al., 2025) and VLM-CoT, of which the training details are elaborated in these works. Additionally, we finetune the VLM in Impromptu (Chi et al., 2025) with CoT reasoning capabilities to enhance the system's interpretability and reasoning performance, particularly for high-complexity scenarios. The finetuning of VLM-CoT is conducted on A 800 GPU. The hyperparameters for finetuning VLM-CoT agent is in the Table 6. The testing of all the models is conducted on 3090 for fair comparison.

For closed-loop evaluation of large models in the CARLA environment, we first generate scenario complexity labels on 50% of the Bench2Drive-Base dataset using the rule-based pipeline described in Section 2.3.1, and train the ScpViT model for scenario complexity prediction on this labeled subset. Then, we follow the testing protocol of Bench2Drive-VL (Thinklab-SJTU, 2025) to conduct closed-loop evaluation. The full test set comprises 220 routes. However, due to the prohibitively high inference cost of large models, we carefully select a representative subset for evaluation, 10 difficult routes (Jia et al., 2024) and 28 routine routes. This subset maintains sufficient challenge to assess model robustness while ensuring evaluation feasibility, effectively capturing the system's overall performance under the long-tailed scenario distribution.

Table 6: Hyperparameters for finetuning VLM-CoT agent.

| Hyperparameter | Value |
| --- | --- |
| Cutoff len | 4096 |
| Finetuning type | full |
| Image resolution | 262144 |
| Learning rate | $5.0 \times 10^{-6}$ |
| Scheduler type | cosine |
| Warmup ratio | 0.03 |

## D    ANNOTATION OF CHAIN-OF-THOUGHT DATA

To enhance the reasoning capability of the Deep Thinking Agent, we fine-tuned the base VLM on a dataset annotated with Chain-of-Thought (CoT) rationales. These CoT annotations were automatically generated using large VLMs. For each scenario, we provided the model with its original annotations, including object categories, velocities, and positions, etc., and a structured three-step reasoning template. The template guided the model to articulate its reasoning process sequentially. The resulting CoT data provides step-by-step rationales to generate the final trajectory, enabling the Deep Thinking Agent to learn an interpretable and safety-aware decision-making process.

## E    MORE EXPERIMENT RESULTS

**Ablation Study on Agent Coordination Strategy.**    We compare TAD against fixed-agent baselines to validate the necessity of dynamic coordination. This proves that the scenario complexity-based dynamic selection mechanism is superior to the "fixed use of the best agent" in balancing performance and resource consumption.

Figure 5: Different scenario complexity levels and example of CoT reasoning for complex driving scenarios.

Table 7: Ablation Study on Agent Coordination Strategy.

| Coordination Strategy | Avg. L2 Error ↓ | Avg. Latency ↓ | Avg. Memory ↓ |
|---|---|---|---|
| Fast Agent Only | 2.0 | 0.22s | 3.4 GB |
| Smart Agent* Only | 0.49 | 5.4s | ≈ 19GB |
| Deep Thinking Agent Only | 0.42 | 5.8s | ≈ 19GB |
| **TAD*(Oracle, 100%)** | 0.47 | 3.8s | ≈ 13.8GB |
| **TAD*(Ours, 92.59%)** | 0.46 | 4.2s | ≈ 15.4GB |

**Validating the Reliability of Scenario Complexity Labeling with Spearman's Rank Correlation Coefficient.** The core utility of our framework lie in the relative ordering of scenarios based on their complexity. Our routing mechanism does not require knowing that a scene has an "exact score of 0.8"; it only needs to know that one scene is more complex than another (e.g., scene A with score 0.8 ¿ scene B with score 0.3).

To rigorously validate that our rule-based labeling scheme captures a meaningful and learnable ordinal relationship, we propose a powerful quantitative metric, Spearman's rank correlation coefficient, $\rho$.

This non-parametric statistic measures the strength and direction of the monotonic relationship between two ranked variables. In our case, the ranking of scenes by our rule-based system versus the ranking predicted by our ScpViT model. The value of $\rho$ ranges from -1 to 1:

- $\rho = 1$: Perfect positive correlation (model's ranking perfectly matches the ground-truth ranking).
- $\rho = 0$: No correlation.
- $\rho = -1$: Perfect negative correlation.

We analyze the monotonic correlation between the scene complexity predicted by SCPViT and the ground truth by calculating Spearman's rank correlation coefficient, $\rho = 0.7740$, p-value $\approx 0.00$. This represents a strong, statistically significant positive correlation, which provides robust evidence for two key claims:

1. The model has successfully learned the core ordinal relationships defined by our rules. ScpViT has learned to predict the relative complexity of scenes in a way that aligns with the human-defined, rule-based hierarchy. This demonstrates that our labeling is not arbitrary but encodes a coherent, learnable structure.

2. The relative complexity ordering captured by our annotation pipeline is meaningful and consistent. The high $\rho$ value proves that the "complexity" we define is not a random construct and it is a signal that can be reliably learned from raw sensor inputs, making it a valid basis for dynamic agent coordination.

In summary, the strong Spearman correlation ($\rho = 0.7740$) provides empirical, statistical validation that our labeling scheme is both meaningful and robust, and that our model has successfully learned

to replicate this critical ordinal structure. p-value $\approx 0.00$ indicates that this correlation is statistically highly significant. This directly addresses the concern about subjectivity and confirms the reliability of our approach.

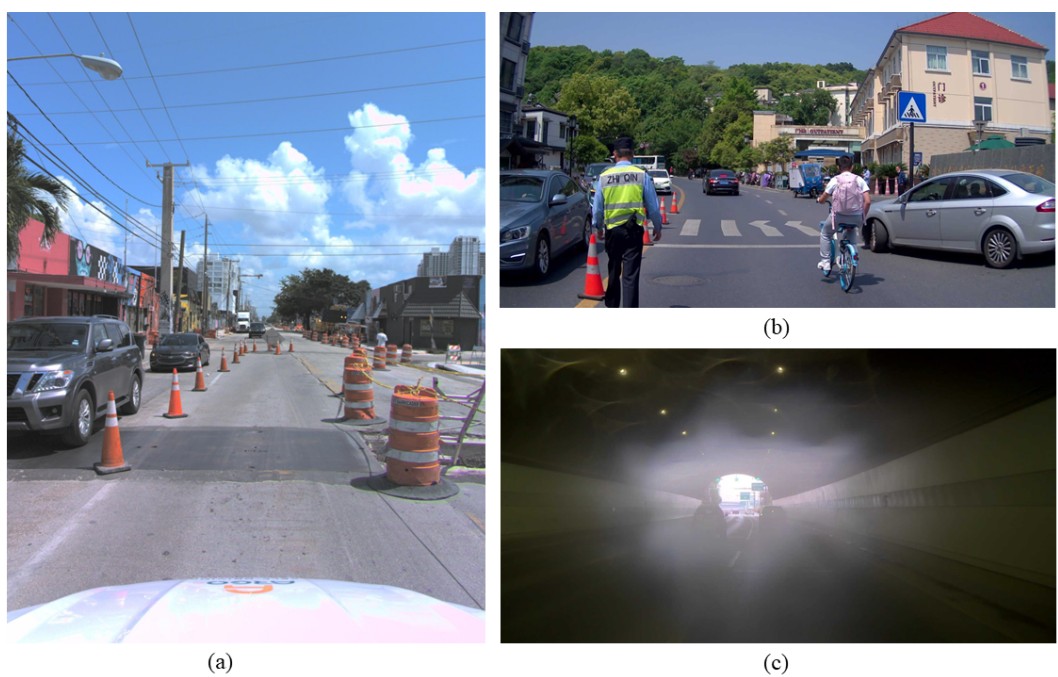

Figure 6: Examples of corner cases in autonomous driving scenarios. (a) Irregular traffic rules. (b) Complex dynamic interactions. (c) Extreme lighting conditions.

**Visualization.** As shown in Fig. 6, We visualize several challenging scenario samples from the dataset, including temporary changes in drivable areas caused by construction zones, complex intersections, driving inside tunnels with abnormal lighting conditions.

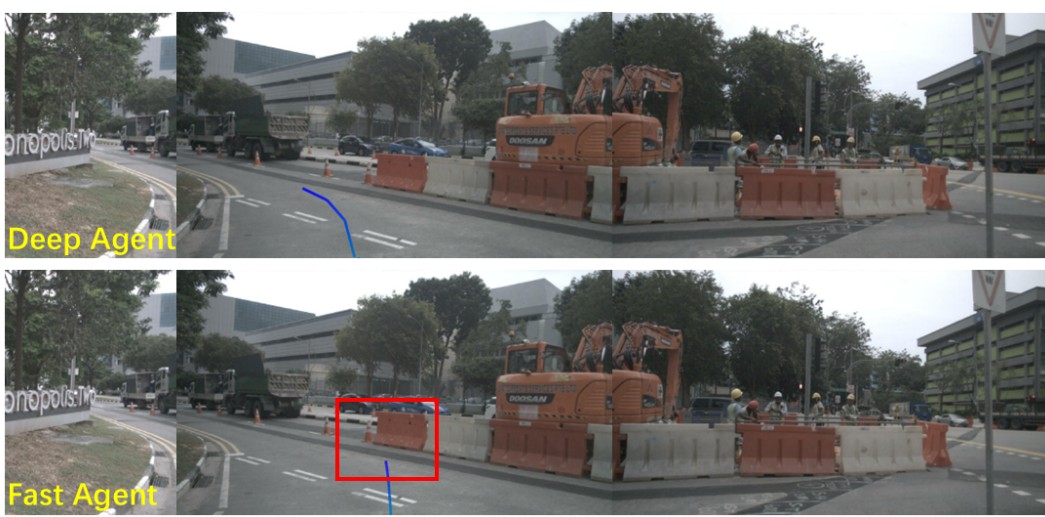

Figure 7: Fast vs Deep Thinking Agent performance in complex scenarios.

As shown in Fig. 7, in complex scenarios, different agents exhibit significant differences in trajectory planning performance. The Fast Agent has limited understanding of the upcoming construction zone, resulting in a planned trajectory that fails to effectively avoid obstacles and carries a risk of colliding with the traffic barriers (water-filled barricades). In contrast, the Deep Agent demonstrates a more accurate perception and semantic understanding of the construction area, generating a trajectory that not only navigates around the obstacles reasonably but also significantly enhances driving safety.

We further make **failure case analysis** on the Agent Coordination module. We find that the ScpViT model exhibits weaker prediction performance in low-complexity scenarios (70.32% in Table 3), often misclassifying them as medium-complexity. Although this misclassification may erroneously trigger the Smart Agent, introducing additional latency, the predicted trajectories remain safe. TAD*(92.59%)'s L2 error is less than TAD*(Oracle, 100%) in Table 7.

In addition, we visualize a typical sample in the Fig. 8. Although this scenario involves only a limited number of target objects, the nighttime setting led the model to misclassify it as medium complexity. Nevertheless, the Smart Agent demonstrates robust performance under such environmental conditions. This also indicates that the model has not effectively disentangled the features of object quantity from environmental context, and instead tends to rely predominantly on the latter for complexity assessment. Future improvements may require strategies that encourage the model to jointly leverage both local and global information for more reliable predictions.

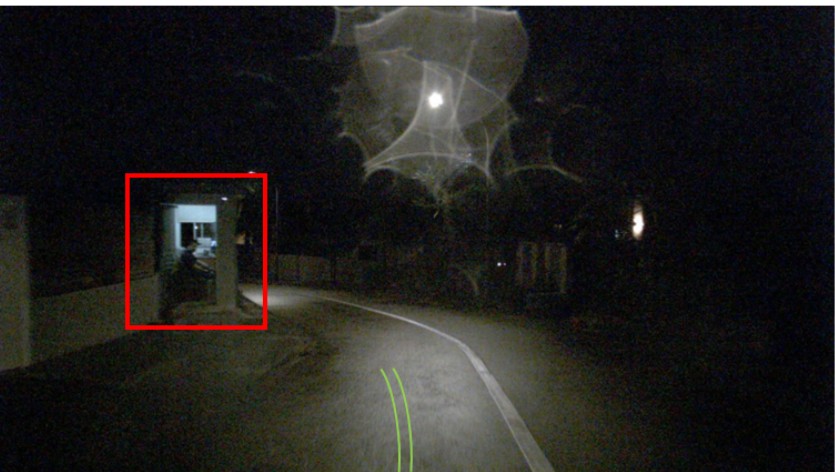

Figure 8: Failure cases.

Moreover, ScpViT performs well on medium-complexity scenarios but shows a tendency to misclassify high-complexity cases as medium-complexity. Fortunately, the Smart Agent possesses a certain degree of fault tolerance. Even when high-complexity situations are underestimated, the Smart Agent can still maintain basic system stability and prevent catastrophic failure. To further enhance the system's robustness and safety, we propose incorporating an uncertainty estimation mechanism: when the prediction uncertainty from ScpViT exceeds a predefined threshold, the system should automatically fall back to the VLM-CoT model, which offers stronger generalization and reasoning capabilities, to provide more reliable decision-making in complex or ambiguous scenarios.

