# OpenReview forum: "Tri-Agent Driving: Learning to Coordinate Agents via Scenario Complexity Representation for Efficient Autonomous Driving"
_ICLR.cc/2026/Conference — Submitted to ICLR 2026_

### Official Review · Reviewer_c1Uy · 2025-10-16

**Soundness:** 3
**Presentation:** 3
**Contribution:** 3
**Rating:** 4
**Confidence:** 5

**Summary:**

This paper proposes a Mix-of-Experts–like planning framework for autonomous driving. In simple scenarios, a faster end-to-end model is used for trajectory planning, while in medium and complex scenarios, a VLM-based large model is employed for scene understanding and planning. The authors introduce a data-driven method for defining scene complexity that does not rely on manually set thresholds, and they design a ViT-based module to assess scene complexity during inference. Experiments conducted on a hybrid dataset fused from multiple open-source datasets demonstrate that the proposed approach achieves consistent advantages over previous strategies.

**Strengths:**

1. The proposed TAD framework presents an intuitive idea: by adopting a Mixture-of-Experts–like approach, it allows the system to use a faster end-to-end model for trajectory planning in most common and simple scenarios, thereby significantly reducing computational cost.
2. The narrative is clear, and the figures and tables are intuitive, making it easy for readers to understand the authors’ methodology and main ideas.

**Weaknesses:**

1. The comparison with Senna in Table 4 is not entirely appropriate, as Senna is a sequential model combining VLM and end-to-end approaches. Its final trajectory is produced by the end-to-end module, rather than being directly generated by the VLM as in the proposed method.
2. The relative definition of scene complexity has certain limitations. For example, in the nuScenes dataset, straight-driving scenarios account for a large portion, so some cases categorized as “complex” may not actually be hard cases. Conversely, if a dataset primarily contains complex scenes, forcing a split into “simple” cases may also be inappropriate.
3. Although the proposed method improves average latency and memory, this metric has limited practical significance, since what matters in real-world applications is the real-time latency.
4. The paper lacks sufficient training details. Specifically, for each agent, is the training performed only on data corresponding to its assigned scene complexity, or on the entire dataset? If it is the former, then misclassification of a complex scene as a simple one could severely degrade performance, as the fast agent may not have been trained on such complex cases.
5. In Table 3, there is a significant performance gap between the fast agent and the other two agents. If this is the case, then regardless of scene complexity, a fully VLM-based model (e.g., Impromptu-3B) should theoretically achieve the best results. However, as shown in Table 4, the proposed hybrid model performs better. Authors should clarify why this is the case.

**Questions:**

1. It is unclear how “Unseen Cases” are labeled and identified. Is this information, such as Unconventional Obstacles, labeled in the dataset?
2. For the scene complexity annotation, were the datasets annotated separately before merging, or was the annotation conducted after merging all original datasets (nuScenes, Impromptu VLA, ...)?
3. Line 357 mentions that the authors further annotated CoT reasoning data. If these annotations were produced using a large model, this should also be explicitly stated in Line 632, where the use of large models is discussed.
4. There seems to be a practical contradiction: complex and safety-critical scenarios, where deep and accurate reasoning is most needed, are often the ones requiring the fastest inference speed.
5. It is recommended to use vector-based PDF figures, as the current rasterized images become blurry when zoomed in.

---

> ### Author Response · Authors · 2025-11-23
> **Response to Weaknesses 1-2**
>
> We sincerely thank the reviewer for their insightful and constructive comments. These suggestions have helped us further clarify the key details in our method design and experimental evaluation. We provide our point-by-point responses as follows.
>
> **1. The comparison with Senna in Table 4 is not entirely appropriate, as Senna is a sequential model combining VLM and end-to-end approaches. Its final trajectory is produced by the end-to-end module, rather than being directly generated by the VLM as in the proposed method.**
>
> We thank the reviewer for this insightful comment. We agree that Senna's architecture employs the VLM to generate meta-actions which are then used by a traditional end-to-end (E2E) module to produce the final trajectory.
>
> **However**, we respectfully argue that the comparison in Table 4 remains valid, because it highlights a fundamental difference in system-level efficiency. The key point is that in Senna, the VLM is an indispensable component in the inference pipeline for every single scenario, regardless of its complexity. This dedsign inherently couples the system's latency and computational cost to the large VLM.
>
> In contrast, our TAD framework introduces a complexity-aware router that dynamically decides whether to invoke a VLM-based agent. For a significant portion of low scenarios, the system operates efficiently using the traditional E2E agent (Fast Agent), completely bypassing the cost of VLM inference. This "fast when possible, deep when necessary" paradigm is the core of our efficiency gains.
>
> Therefore, while Senna sequentially fuses VLM outputs into an E2E pipeline, its efficiency is perpetually bounded by the VLM's overhead. Our method, by dynamically routing around this overhead in simple cases, achieves a superior system-level trade-off, as empirically demonstrated by the significantly lower average latency and memory consumption in Table 4.
>
> **2. The relative definition of scene complexity has certain limitations. For example, in the nuScenes dataset, straight-driving scenarios account for a large portion, so some cases categorized as “complex” may not actually be hard cases. Conversely, if a dataset primarily contains complex scenes, forcing a split into “simple” cases may also be inappropriate.**
>
> We acknowledge that defining scenario complexity based on a single dataset may introduce limitations due to potential biases in data distribution. To address this concern, we did not rely solely on the nuScenes dataset but instead constructed a hybrid benchmark integrating multiple datasets including NuScenes, NuScenes-C, and Impromptu. The complexity thresholds were determined based on the overall distribution of this combined dataset. This design aims to comprehensively cover diverse scenarios ranging from regular cases to extreme edge cases, significantly reducing model bias caused by limited training data sources.
>
> To further enhance the system's robustness in real-world deployment, we have incorporated crucial safety fallback mechanism into the architecture. We recognize that learning-based routing modules inevitably carry predictive uncertainties. Therefore, when the module's output uncertainty exceeds a predefined threshold, the system automatically switches to the most capable "Deep Thinking Agent". This mechanism effectively addresses two types of misjudgments: if complex scenarios are incorrectly classified as simple, the powerful agent provides enhanced safety assurance; even if simple scenarios are misclassified as complex, the Deep Thinking Agent still maintains superior performance while only incurring minimal efficiency costs,  ensuring overall system safety is consistently preserved.
>
> We enhance TAD* with an entropy-based uncertainty estimation mechanism: when prediction uncertainty is high, the system falls back to the Deep Agent for safe inference.  As shown in the table, this strategy reduces the average L2 error from 0.46 to 0.40 (a 13% improvement) with only a modest increase in latency (+0.6s) and memory usage (+1.4GB), effectively balancing safety and accuracy.
>
> |                    | Avg. L2 Error ↓ | Avg. Latency ↓ | Avg. Memory ↓ |
> | ------------------ | --------------- | -------------- | ------------- |
> | TAD*(Ours, 92.59%) | 0.46            | 4.2s           | ≈ 15.4GB      |
> | TAD*+  Uncertainty |       0.40         |      4.8s |      ≈ 16.8GB         |

---

> ### Author Response · Authors · 2025-11-23
> **Response to Weaknesses 3-5**
>
> **3. Although the proposed method improves average latency and memory, this metric has limited practical significance, since what matters in real-world applications is the real-time latency.**
>
> The "End-to-End (E2E) inference latency" reported in Table 4 was measured in a laboratory environment without applying common engineering optimizations used in real-world deployments. In an actual vehicle system, latency could be further substantially reduced through strategies such as model quantization, or distillation, to meet the real-time requirements of autonomous driving.
>
> The core contribution of this work lies in demonstrating that  the proposed dynamic multi-agent architecture can significantly reduce the system’s average computational cost and GPU memory consumption without compromising overall performance. This design is inherently extensible and holds strong optimization potential. When deployed in real vehicle systems, it is expected to yield a system-wide reduction in resource usage, offering a practical pathway toward building efficient and adaptive autonomous driving systems.
>
> **4. The paper lacks sufficient training details. Specifically, for each agent, is the training performed only on data corresponding to its assigned scene complexity, or on the entire dataset? If it is the former, then misclassification of a complex scene as a simple one could severely degrade performance, as the fast agent may not have been trained on such complex cases.**
>
> The training details are as follows:
>
> Fast Agent. We adopt VAD, which is trained on the nuScenes dataset. NuScenes includes diverse real-world driving conditions such as varying weather, illumination, and traffic density.
>
> Smart Agent. The Smart Agent is based on the model trained on Impromptu dataset from [1], enabling robust generalization across different complexity levels.
>
> Deep Thinking Agent. Based on the Smart Agent, we further finetune it using our CoT-annotated data specifically designed for high-complexity corner cases.
>
> The key contribution of this work lies not in pursuing performance breakthroughs with a single agent model, but in proposing a scenario complexity-aware dynamic driving system with multi agents. By introducing the lightweight ScpViT module for real-time scenario complexity prediction and routing decisions, the system can dynamically select the most suitable driving agent for the current scenario, achieving an optimal balance between efficiency and accuracy at the overall system level.
>
> To further enhance the system's robustness in practical deployment, we have incorporated critical safety fallback mechanism into the architecture. This mechanism effectively addresses two types of misclassification scenarios: if complex situations are misjudged as simple, the powerful agent provides enhanced safety guarantees; even if simple scenarios are incorrectly classified as complex, the Deep Thinking Agent maintains superior performance with only minor sacrifices in inference efficiency, thereby ensuring overall system safety is consistently preserved.
>
> **5. In Table 3, there is a significant performance gap between the fast agent and the other two agents. If this is the case, then regardless of scene complexity, a fully VLM-based model (e.g., Impromptu-3B) should theoretically achieve the best results. However, as shown in Table 4, the proposed hybrid model performs better. Authors should clarify why this is the case.**
>
> Table 4 reports the overall performance of models on the hybrid test set (Avg. L2 Error), which is essentially a weighted average, with weights corresponding to the proportions of scenarios at different complexity levels in the test set. TAD* achieves a strategic trade-off across different scenarios:
>
> (1) In low-complexity scenarios, the Fast Agent exhibits higher error compared to the Smart Agent*. However, this minor peformance sacrifice is well justified by the dramatic gain in efficiency, reducing inference latency from 5.4 s to just 0.22 s.
>
> (2) In medium- and high-complexity scenarios, TAD* correctly routes inputs to the Smart Agent* or the Deep Thinking Agent, respectively, maintaining strong performance where it matters most.
>
> As a result, TAD drastically reduces average system latency and GPU memory consumption by routing these simple cases to the highly efficient Fast Agent. Meanwhile, by invoking powerful models (CoT model) for complex scenes, TAD preserves, or even slightly improves, overall trajectory prediction accuracy (Avg. L2 Error), as computational resources are allocated more effectively across the long-tailed distribution of driving scenarios.

---

> ### Author Response · Authors · 2025-11-23
> **Response to Question 1-5**
>
> **1. It is unclear how “Unseen Cases” are labeled and identified. Is this information, such as Unconventional Obstacles, labeled in the dataset?**
>
> We thank the reviewer for this question, which helps us clarify an important point. The term "Unseen Cases" refers to what is more commonly known as "Corner Cases" characterized by distribution shifts. We will revise the manuscript to use this more precise terminology to avoid confusion. The corner cases include Unconventional Obstacles , Traffic Rule Changes , and Challenging Road Conditions from Impromptu [1].
>
> [1] Haohan Chi, Huan-ang Gao, Ziming Liu, Jianing Liu, Chenyu Liu, Jinwei Li, Kaisen Yang, Yangcheng Yu, Zeda Wang, Wenyi Li, et al. Impromptu vla: Open weights and open data for driving vision-language-action models. NeurIPS 2025.
>
> **2. For the scene complexity annotation, were the datasets annotated separately before merging, or was the annotation conducted after merging all original datasets (nuScenes, Impromptu VLA, ...)?**
>
> We constructed a hybrid benchmark integrating multiple datasets including NuScenes, NuScenes-C, and Impromptu. The complexity thresholds were determined based on the overall distribution of this combined dataset. This design aims to comprehensively cover diverse scenarios ranging from regular cases to extreme edge cases, significantly reducing model bias caused by limited training data sources.
>
> **3. Line 357 mentions that the authors further annotated CoT reasoning data. If these annotations were produced using a large model, this should also be explicitly stated in Line 632, where the use of large models is discussed.**
>
> We thank the reviewer for this point. In response, we have revised the manuscript to explicitly state the data generation process. The CoT annotations were generated using large models, guided by a structured three-step template and based on the original annotations. A corresponding statement has also been added to our disclosure regarding the use of large models, clarifying that its use was confined to the creation of CoT training labels.
>
> **4. There seems to be a practical contradiction: complex and safety-critical scenarios, where deep and accurate reasoning is most needed, are often the ones requiring the fastest inference speed.**
>
> We thank the reviewer for this insightful observation! There indeed exists a fundamental tension between "deep reasoning" and "extremely fast response" in autonomous driving systems, a contradiction that is difficult to be resolved within a single-model architecture. The core innovation of  TAD lies in transforming this contradiction into an advantage of "on-demand allocation" through system-level intelligent coordination. Specifically:
>
> For most high-complexity scenarios (e.g., unprotected left turns,  complex traffic participant intentions), the challenge lies in semantic comprehension and strategic reliability, not instantaneous reaction. The decision-making time window for such scenarios is more relaxed, where safety relies on making correct and reliable decisions rather than achieving ultimate speed. TAD proactively activates the Deep Thinking Agent through scenario complexity prediction, trading a manageable latency cost (as shown in Table 4) for significantly enhanced decision-making safety. In these scenarios, a quick but error trajectory is far more dangerous than a slightly delayed but correct one.
>
> For truly critical scenarios requiring millisecond-level responses (e.g., imminent collisions), any "deep reasoning" reliant on large models falls outside the effective response window. In these cases, system safety is guaranteed by the low-latency Fast Agent and underlying safety controllers (e.g., emergency braking modules).
>
> Therefore, the true value of TAD is its ability to precisely deploy expensive deep reasoning capabilities in scenarios that "require intelligence but allow time for thinking," while substantially conserving computational resources in simple scenarios. Consequently, at the system level, it achieves safety comparable to persistently using large models, while reducing average inference latency by 26% and GPU memory consumption by 30% (as demonstrated in Table 4).
>
> **5. It is recommended to use vector-based PDF figures, as the current rasterized images become blurry when zoomed in.**
> Thank you very much for your suggestion. We will include figures in PDF format in the revised manuscript.

---

> > ### Comment · Reviewer_c1Uy · 2025-11-26
> >
> > Thank you for the detailed responses, which addressed most of my questions regarding the model details and experiements. However, I think the current paper still has the following key issues:
> >
> > - The paper emphasizes that the proposed method reduces average inference latency and memory usage. Yet in real-world applications, it is the peak latency that truly matters for real-time systems. Reducing the average latency only implies lower overall computation, but does not relax the hardware requirements: the system must still provision enough computational resources at all times, since it is unclear when the large model will be needed.
> > - The core of the method is essentially an MoE architecture combining an end-to-end “fast” system and a VLM-based “slow” system, with a ViT-based scene classifier dynamically selecting which planner to use. However, scene complexity classification is inherently ambiguous, and the fact that different models are trained on different datasets may lead to inconsistencies in planning when switching between models.
> > - I think the proposed method should be compared with other state-of-the-art algorithms on more widely used public benchmarks, such as Bench2Drive and NAVSIM. Currently, whether it is open-loop or closed-loop, the results of the comparison algorithms seem to be mostly the results reproduced by the authors rather than the original results of other papers.
> >
> > For these reasons, I am inclined to maintain my original score.

---

> > > ### Author Response · Authors · 2025-11-28
> > >
> > > Thank you for your prompt response. We wish to provide further clarification on the following points:
> > >
> > > 1.	Regarding TAD's Core Design Objectives and Contributions
> > >
> > > The primary design goal of TAD is not to directly reduce the peak latency or absolute hardware requirements of large VLMs. We fully recognize that deploying large VLM models onto vehicles is another significant challenge, involving underlying technologies such as model compression.
> > >
> > > The core issue TAD aims to address stems from the long-tail distribution characteristics of real-world autonomous driving scenarios. Within this distribution, continuously invoking large models to handle a vast number of simple, routine scenarios leads to significant computational resource wastage. Therefore, TAD’s key contribution lies in proposing a dynamic computational resource allocation architecture. Its core idea is: on a given hardware platform capable of running the most complex models, maximize the efficiency of computational resource utilization through intelligent scheduling.
> > >
> > > 2.	Regarding the Reliability, Ambiguity, and Safety of Scene Classification
> > >
> > > First, our approach does not pursue absolutely correct scene classification but relies on the relative ordering of scene complexity. The Spearman’s rank correlation coefficient (ρ = 0.7740, p ≈ 0.0000) strongly demonstrates that our annotation method produces meaningful, learnable ordinal signals, which are sufficient to serve as a reliable basis for routing decisions.
> > > Through an uncertainty-aware fallback mechanism, we transform the inherent ambiguity of classification into a safety feature: when the predictor is uncertain, the system automatically activates the most capable "deep-thinking agent," ensuring safety priority in edge cases.
> > >
> > > Second, although the three agents in the framework differ in architecture and training data, their output spaces are unified (all generating physically feasible trajectories), and their optimization objectives are consistent.
> > >
> > > 3.	Regarding the Paper’s Core Contribution and Experimental Evaluation
> > >
> > > We wish to emphasize that the core contribution of this paper is not proposing a new, single SOTA model, but rather introducing a dynamic resource allocation framework. Although the baseline results stem from our reproduction, the conclusions regarding "efficiency improvement" remain solid and compelling. Following your suggestion, we will endeavor to supplement the comparative analysis with official results from the latest SOTA methods on the complete Bench2Drive and NAVSIM benchmarks, which will be an important step to further strengthen this work.
> > >
> > > In conclusion, the essence of the TAD framework is a dynamic scheduling system centered on optimizing computational resource utilization efficiency, rather than a new model aimed at achieving lower latency or smaller model size. It aims to address the challenges of long-tail scenarios and scene classification ambiguity through reliable relative complexity ordering and a fallback mechanism that transforms uncertainty into a safety guarantee.

---

### Official Review · Reviewer_uWbS · 2025-10-27

**Soundness:** 2
**Presentation:** 3
**Contribution:** 2
**Rating:** 2
**Confidence:** 4

**Summary:**

The paper introduces Tri-Agent Driving (TAD), a hierarchical autonomous driving framework with three agents, that handle scenes of increasing complexity.
A complexity-based routing module classifies each scene into easy, normal, or hard levels to decide which agent plans the trajectory.
TAD aims to balance accuracy, latency, and computational cost, achieving comparable planning performance with lower delay and memory than single-agent or VLM baselines.
The main contributions are the complexity-driven routing strategy, and analysis showing improved efficiency trade-offs.

**Strengths:**

1. The TAD architecture cleanly decomposes tasks by scene complexity, making the system easy to reason about and extend.
2. Complexity-based routing allocates compute where needed, yielding better latency/memory trade-offs than a single heavy model.
3. The paper is well organized and written in a clear, easy-to-follow manner.

**Weaknesses:**

1. Lack of empirical analysis of existing hybrid systems. Although the authors claim in the related work section that pipeline of DriveVLM & Senna (treating VLM outputs as fixed auxiliary inputs to downstream modules) is suboptimal, this limitation is not empirically validated. Presenting comparative experiments to demonstrate such drawbacks would make the motivation for introducing TAD much more convincing.
2. Questionable validity of complexity labels. The complexity labels used to decide which agent handles planning are defined purely from a scene-modeling perspective and are hard-coded using the $<Q1$ and $>Q3$ thresholds. This approach seems arbitrary, lacking empirical analysis.
3. Limited scenario complexity in the dataset. The dataset used does not appear to include highly challenging or diverse driving scenes, which restricts the evaluation of the proposed system’s robustness and scalability.
4. Absence of closed-loop evaluation. The results are primarily based on open-loop metrics such as L2 trajectory error. Without closed-loop testing, it is difficult to assess performance, especially the paper aims to address some complex scenarios.

**Questions:**

1. Does the system include any mechanism to estimate uncertainty or fallback when the complexity classifier is uncertain or misclassifies a scene? This could be critical for safety in deployment.
2. You mention that existing hybrid systems (DriveVLM & Senna), where VLM outputs are treated as fixed auxiliary inputs, have notable limitations. Could you provide empirical evidence or ablation studies demonstrating this weakness to strengthen your motivation for TAD?

---

> ### Author Response · Authors · 2025-11-23
> **Response 1**
>
> We thank Reviewer for the detailed feedback and positive comments that the paper is **well organized and written in a clear, easy-to-follow manner**. We have addressed all concerns below to further support our claims.
>
>  **W1: Lack of empirical analysis of existing hybrid systems.**  **Q2: Could you provide empirical evidence or ablation studies demonstrating this weakness to strengthen your motivation for TAD?**
>
> For serialized architectures like Senna, the overall inference latency is determined by two sequentially executed stages: the traditional E2E model and the large model, both of which must run consecutively. In contrast, TAD adopts a parallelized module design: for each input sample, the system only needs to invoke either fast agent or the large model for inference, rather than executing both. Consequently, on the same fixed test set, since some samples can be processed solely by the lightweight agent, TAD significantly outperforms serialized architectures in terms of both average inference time and GPU memory. As shown in Figure 1 (f) in the manuscript, on a challenging hybrid test set spanning diverse traffic conditions, TAD achieves state-of-the-art trajectory prediction, while reducing average inference latency by 26\% (4.2s vs. 5.7s) and GPU memory consumption by 30\% (15.4 GB vs. 22 GB) compared to Senna.
>
> **W2: Questionable validity of complexity labels.This approach seems arbitrary, lacking empirical analysis.**
>
> We thank the reviewer for raising these points about the foundation of our complexity labels. We would like to clarify that our approach is not arbitrary but is a rule-based, systematic pipeline designed to translate established domain challenges into a quantifiable metric. Our scenario complexity is defined through a interpretable, rule-based system grounded in domain knowledge and data-driven statistics.
>
> **Firstly,** the values and complexity indices in Table 1 were not chosen arbitrarily. They were assigned based on well-established challenges in the autonomous driving literature[2,3,4]:
>
> Environmental Conditions. It is a consensus that perception performance degrades under adverse conditions. Thus, we applied a rule: clear weather (0.0) < rain (0.5) < fog/snow (1.0), reflecting the increasing level of perceptual difficulty.
>
> Traffic Density. A higher number of interactive agents unequivocally increases planning complexity. Our assignment follows a simple, monotonically increasing rule based on the number of bounding boxes and participant types.
>
> Unseen Cases. Scenarios involving distribution shifts (e.g., unconventional obstacles) are widely recognized as the most challenging for generalization. We applied a binary rule: if present, assign the maximum complexity (1.0) as they necessitate the highest level of caution and reasoning.
>
> **Secondly,**  regarding the complexity level partitioning, the use of the 25th and 75th percentiles (Q1, Q3) is a data-driven and statistically principled choice, not a hard-coded one.
>
> This method is based on the Interquartile Range (IQR)[1], a robust measure of statistical dispersion. It automatically adapts to the distribution of any given dataset, making it more generalizable than fixed thresholds that would be dataset-specific.
>
> The resulting three regions,'Low' (bottom 25%), 'Medium' (middle 50%), and 'High' (top 25%), provide an intuitive and balanced partitioning of scenarios into "easier-than-average," "average," and "harder-than-average" categories. This is a common and effective practice in data analysis for creating balanced splits.
>
> **Thirdly**, the most compelling justification for our rule-based complexity pipeline is its empirical validity in the downstream autonomous driving task. The significant performance gains of the full TAD system (Table 4) serve as strong evidence. If our complexity labels were invalid or meaningless, the dynamically coordinated system would not consistently and significantly outperform all static baselines, including the "best agent only" strategy.
>
> **More importantly**, we further compute the Spearman’s rank correlation coefficient as a core empirical validation. This provides strong evidence that the relative complexity ranking captured by our annotation scheme is meaningful and can be consistently learned, rather than being based on arbitrary human subjectivity.
>
> [1] John Wilder Tukey et al. Exploratory data analysis, volume 2. Springer,1977.
>
> [2] J. Wang, C. Zhang, Y. Liu and Q. Zhang, "Traffic Sensory Data Classification by Quantifying Scenario Complexity," 2018 IEEE Intelligent Vehicles Symposium (IV), 2018, pp. 1543-1548.
>
> [3] RoboBEV: Towards Robust Bird's Eye View Perception under Corruptions. S Xie, L Kong, W Zhang, J Ren, L Pan, K Chen, Z Liu. TPAMI 2025.
>
> [4] Haohan Chi, Huan-ang Gao, Ziming Liu, Jianing Liu, Chenyu Liu, Jinwei Li, Kaisen Yang, Yangcheng Yu, Zeda Wang, Wenyi Li, et al. Impromptu vla: Open weights and open data for driving vision-language-action models. NeurIPS 2025.

---

> ### Author Response · Authors · 2025-11-23
> **Response 2**
>
> **Weakness 3: Limited scenario complexity in the dataset. The dataset used does not appear to include highly challenging or diverse driving scenes, which restricts the evaluation of the proposed system’s robustness and scalability.**
>
> We would like to clarify that our hybrid test set is specifically designed to include a wide spectrum of challenging and diverse driving scenes.
>
> **Firstly**, our dataset is not built upon a single source. It is a hybrid set that integrates:
>
> The standard nuScenes dataset for common scenarios.
>
> The nuScenes-C benchmark, which is explicitly designed to test robustness against real-world corruptions and challenging environmental conditions.[1]
>
> The Impromptu dataset, a recently released and publicly available benchmark that is specifically curated for long-tailed and corner-case scenarios in autonomous driving.[2]
>
> **Secondly,** as shown in Figure 4, the scenario distribution in our dataset demonstrates a significant long-tail characteristic. This distribution pattern highly aligns with the complex composition of real-world driving scenarios, indicating that our evaluation comprehensively covers a wide spectrum of situations, from common cases to rare corner cases. We have included example images of several corner cases in the Appendix of the manuscript.
>
> **Thirdly,** the superior performance of our Deep Thinking Agent in high-complexity scenarios (Table 3) and the overall efficiency gains of the TAD system (Table 4) were demonstrated precisely on this hybrid set. The fact that our method shows significant improvement, especially in handling the complex tail of the distribution, provides strong evidence for its robustness and scalability.
>
> [1] RoboBEV: Towards Robust Bird's Eye View Perception under Corruptions. S Xie, L Kong, W Zhang, J Ren, L Pan, K Chen, Z Liu. TPAMI 2025.
>
> [2] Haohan Chi, Huan-ang Gao, Ziming Liu, Jianing Liu, Chenyu Liu, Jinwei Li, Kaisen Yang,
> Yangcheng Yu, Zeda Wang, Wenyi Li, et al. Impromptu vla: Open weights and open data for
> driving vision-language-action models. NeurIPS 2025.
>
> **Weakness 4:  Absence of closed-loop evaluation. The results are primarily based on open-loop metrics such as L2 trajectory error. Without closed-loop testing, it is difficult to assess performance, especially the paper aims to address some complex scenarios.**
>
> We incorporate multiple baseline methods and conduct closed-loop experiments on CARLA, evaluated with more comprehensive metrics. The results are presented below (an 80s early stop is used, so the success rate is lower than normal evaluation methods). As shown in the table, TAD achieves an RC of 64.34, representing a 4.83 percentage point improvement over the strongest baseline model, Impromptu and its Driving Score (DS) is 42.50, also higher (+0.67).  For more experimental details, please refer to the revised manuscript.
>
> | Method               | RC ↑  | DS   ↑ |
> | -------------------- | ----- | ------ |
> | AD-MLP               | 0.00  | 13.05  |
> | UniAD-Base           | 51.81 | 36.12  |
> | VAD-Base             | 48.48 | 34.66  |
> | Qwen3-2B             | 58.69 | 40.70  |
> | Impromtu(Qwen2.5-3B) | 59.51 | 41.83  |
> | TAD(ours)            | 64.34 | 42.50  |
>
> **Metrics.** Two major metrics: route completion (RC), and driving score (DS). The route completion refers to the percentage of the total route length that has been completed. It only takes into account the distance traveled along the predetermined route, where each segment of the predetermined route corresponds to a navigation instruction. If the agent deviates too far from the route, the agent is regarded as violating the instruction, and this episode is marked as a failure and terminated.  The driving score is the product of the route completion ratio and the infraction score, describing both driving progress and safety.
>
> **Qeustion1: Does the system include any mechanism to estimate uncertainty or fallback when the complexity classifier is uncertain or misclassifies a scene? This could be critical for safety in deployment.**
>
> |                    | Avg. L2 Error ↓ | Avg. Latency ↓ | Avg. Memory ↓ |
> | ------------------ | --------------- | -------------- | ------------- |
> | TAD*(Ours, 92.59%) | 0.46            | 4.2s           | ≈ 15.4GB      |
> | TAD*+  Uncertainty |       0.40         |      4.8s |      ≈ 16.8GB         |
>
> We enhance TAD* with an entropy-based uncertainty estimation mechanism: when prediction uncertainty is high, the system falls back to the Deep Agent for safe inference. As shown in the table, this strategy reduces the average L2 error from 0.46 to 0.40 (a 13% improvement) with only a modest increase in latency (+0.6s) and memory usage (+1.4GB), effectively balancing safety and accuracy.

---

### Official Review · Reviewer_VAfo · 2025-10-31

**Soundness:** 2
**Presentation:** 2
**Contribution:** 2
**Rating:** 2
**Confidence:** 5

**Summary:**

This manuscript introduces Tri-Agent Driving, a framework that dynamically routes multi-view sensor inputs to one of three agents: a lightweight Fast Agent, a medium-capacity Smart Agent, and a CoT-enhanced Deep Thinking Agent,  based on a learned scenario-complexity signal (ScpViT). The idea of adaptive, on-demand activation of differently sized reasoning modules is compelling and addresses an important deployment trade-off: conserve compute in routine cases while invoking heavier reasoning for long-tail corner cases.

**Strengths:**

The paper tackles an important topic: how to allocate computation adaptively in end-to-end autonomous driving systems. The high-level idea of dynamic routing among multiple reasoning agents is appealing, and the motivation to balance efficiency and robustness is timely and relevant to practical deployment

**Weaknesses:**

However, in its current form, the paper has substantial methodological and empirical weaknesses that limit confidence in its claims.

Abstract:
The abstract does not clearly communicate what problem is being solved or why it matters. Even after reading it several times, I still could not pinpoint the exact task or contribution. The writing feels vague and overly general, lacking concrete details that help readers understand the core objective and novelty.

Intro:
The introduction never explicitly defines the research problem. It repeatedly emphasizes “trade-offs” between computation and reasoning, but does not specify for which task or under what setting. I only realized on the second page that the paper targets end-to-end autonomous driving, which should have been stated up front.

Additionally, the mention of "long-tail" scenarios (line 84) requires further justification. What constitutes a long-tail scenario in the context of autonomous driving? How is it formally defined, and what is the data distribution used in the experiments? The introduction also relies on overly vague and promotional phrasing, which should be revised for clarity and focus.

Finally, the claimed contribution regarding speed improvement lacks essential experimental details, such as the hardware setup or computational environment, making it difficult to assess its validity.

Methodology：
The methodological presentation suffers from a lack of logical flow. If the goal is to address an end-to-end task, the section should begin with a clear problem definition and necessary preliminaries. Instead, the authors jump directly into the model architecture, leaving readers confused about the overall setup.

I have serious concerns regarding the proposed scene complexity computation. How are factors such as “Environment,” “Entities,” and “unseen cases” actually derived? Are they obtained from front-view images or historical agent states? The process for identifying elements like “ambiguous traffic signs” or “moderate pedestrian density” from limited input data remains unexplained. The inputs and outputs of the model are not clearly defined, making the entire pipeline difficult to follow.

Moreover, the manual assignment of values and complexity indices for different factors appears arbitrary. The authors do not justify how these values were chosen or whether they generalize across different scenarios. The criteria for complexity level划分 also lack theoretical or empirical support.

The core contribution appears to be ScpViT, but the design of the complexity computation seems relatively straightforward and lacks clear innovation. The remaining components (e.g., VLA and CoT modules) are adapted from existing works, which further diminishes the novelty of the approach. More importantly, the implementation details of these modules are not sufficiently elaborated.

Experiments：
The experimental section has several critical shortcomings. The custom dataset appears to be filtered according to the model’s own complexity classification, which introduces potential bias and raises concerns about circular evaluation.

The experiments are conducted only on this custom dataset without validation on established public benchmarks, which limits the credibility of the reported results. Furthermore, the selected baselines are outdated—mostly from two or three years ago—and do not reflect the current state of the art. The number of baselines is also limited, weakening the comparative analysis.

Most tests are performed in open-loop settings, which are insufficient for validating model effectiveness in real-world driving. Closed-loop evaluation following standard protocols would be more appropriate. The inclusion of trajectory prediction metrics also seems inconsistent with the initial end-to-end framing; if trajectory prediction is a focus, relevant baselines and comprehensive metrics from that subfield should be included.

As someone familiar with this area, I believe the authors should deepen their understanding of autonomous driving tasks and evaluation methodologies before revising. In its current form, the manuscript contains substantial flaws in motivation, method design, and experimental validation, and I do not believe it meets the bar for publication at ICLR.

**Questions:**

See the content in the Weaknesses section.

---

> ### Author Response · Authors · 2025-11-23
> **Response 1 to writing issues**
>
> We thank the Reviewer for the detailed feedback and positive comments, particularly the recognition that our paper **tackles an important topic and that the high-level idea of dynamic routing is appealing**. We believe the primary concerns raised stem from writing issues and certain misunderstandings regarding the proposed scene complexity computation. We have addressed all concerns below to further support our claims.
>
> We appreciate the reviewer's emphasis on clarity and have revised the manuscript accordingly to enhance readability. Specifically, we have **improved the abstract and introduction to provide upfront clarity for all readers**. In the revised introduction, we now begin by explicitly highlighting the long-tailed distribution challenge in real-world end-to-end autonomous driving, where most scenarios are routine and simple, while a small fraction of rare, high-complexity scenarios (such as semantic ambiguity, sudden lane-cutting, and low-light conditions) account for the majority of system failures. This phenomenon necessitates a system that operates efficiently in routine situations while being capable of invoking powerful reasoning when needed. This is a balance that existing methods fail to achieve, as they either prioritize efficiency with poor generalization or offer strong semantics at prohibitive computational cost. Furthermore, we have **revised the Methodology section with a clearer problem formulation**.
>
> These revisions ensure the manuscript more effectively communicates the motivation and value of our approach. Please refer to the revised version (that has been uploaded).
>
> Additionally, we would like to clarify two specific details:
>
> 1. Regarding long-tail scenarios, we refer to rare yet critical driving situations, such as unexpected road obstacles, that occur infrequently in standard driving datasets yet significantly impact safety. The dataset used in our experiments exhibits a long-tailed distribution overall. The specific distribution and scenario categories are presented in Figure 4 (a) of the main manuscript.
>
> 2. All experiments were conducted under identical hardware configurations to ensure fair comparison. Detailed experimental settings are provided in the Appendix C of the manuscript.

---

> ### Author Response · Authors · 2025-11-23
> **Response 2 to the misunderstanding of  scene complexity computation**
>
> **1. Regarding the proposed scene complexity computation**
>
> The prediction of scenario complexity  involves two distinct stages:
>
> **Stage 1 (Offline Annotation)** a transparent, rule-based pipeline that leverages the existing rich annotations in the dataset to generate reliable ground-truth complexity labels for each scenario.
>
> **Stage 2 (Model Training)** a lightweight network, ScpViT, is trained. It takes surround-view images as input and utilizes the labels generated in Stage 1 for supervised learning, achieving a mapping from pure visual input to complexity levels.
>
> **The specific process for the Stage 1 offline annotation is as follows:**
>
> 1. Existing Dataset Annotations. The standard structured annotations corresponding to these images, provided by the autonomous driving dataset. These include 2D/3D object bounding boxes (for vehicles, pedestrians, etc.), object categories,  attribute labels (e.g., type, ID), scene-level metadata (e.g., timestamp, location, labels for weather and illumination).
>
> 2. Label Calculation. Using the information above, the complexity label for each scenario is computed according to the rules specified in Table 1, the calculation method defined in Equation 1, and the final classification label is achieved via the Dual-Threshold Mechanism poposed in the paper.
>
> **2. The manual assignment of values and complexity indices for different factors appears arbitrary.**
>
> We would like to clarify that our approach is not arbitrary but is a rule-based, systematic pipeline designed to translate established domain challenges into a quantifiable metric. Our scenario complexity is defined through a interpretable, rule-based system grounded in domain knowledge and data-driven statistics.
>
> **Firstly,** the values and complexity indices in Table 1 were not chosen arbitrarily. They were assigned based on well-established challenges in the autonomous driving literature[2,3,4]:
>
> Environmental Conditions. It is a consensus that perception performance degrades under adverse conditions. Thus, we applied a rule: clear weather (0.0) < rain (0.5) < fog/snow (1.0), reflecting the increasing level of perceptual difficulty.
>
> Traffic Density. A higher number of interactive agents unequivocally increases planning complexity. Our assignment follows a simple, monotonically increasing rule based on the number of bounding boxes and participant types.
>
> Unseen Cases. Scenarios involving distribution shifts (e.g., unconventional obstacles) are widely recognized as the most challenging for generalization. We applied a binary rule: if present, assign the maximum complexity (1.0) as they necessitate the highest level of caution and reasoning.
>
> **Secondly,**  regarding the complexity level partitioning, the use of the 25th and 75th percentiles (Q1, Q3) is a data-driven and statistically principled choice, not a hard-coded one.
>
> This method is based on the Interquartile Range (IQR)[1], a robust measure of statistical dispersion. It automatically adapts to the distribution of any given dataset, making it more generalizable than fixed thresholds that would be dataset-specific.
>
> The resulting three region, 'Low' (bottom 25%), 'Medium' (middle 50%), and 'High' (top 25%), provide an intuitive and balanced partitioning of scenarios into "easier-than-average," "average," and "harder-than-average" categories. This is a common and effective practice in data analysis for creating balanced splits.
>
> **Thirdly**, the most compelling justification for our rule-based complexity pipeline is its empirical validity in the downstream autonomous driving task. The significant performance gains of the full TAD system (Table 4) serve as strong evidence. If our complexity labels were invalid or meaningless, the dynamically coordinated system would not consistently and significantly outperform all static baselines, including the "best agent only" strategy.
>
> **More importantly**, we further compute the Spearman’s rank correlation coefficient as a core empirical validation. This provides strong evidence that the relative complexity ranking captured by our annotation scheme is meaningful and can be consistently learned, rather than being based on arbitrary human subjectivity.
>
> [1] John Wilder Tukey et al. Exploratory data analysis, volume 2. Springer,1977.
>
> [2] J. Wang, C. Zhang, Y. Liu and Q. Zhang, "Traffic Sensory Data Classification by Quantifying Scenario Complexity," 2018 IEEE Intelligent Vehicles Symposium (IV), 2018, pp. 1543-1548.
>
> [3] RoboBEV: Towards Robust Bird's Eye View Perception under Corruptions. S Xie, L Kong, W Zhang, J Ren, L Pan, K Chen, Z Liu. TPAMI 2025.
>
> [4] Haohan Chi, Huan-ang Gao, Ziming Liu, Jianing Liu, Chenyu Liu, Jinwei Li, Kaisen Yang, Yangcheng Yu, Zeda Wang, Wenyi Li, et al. Impromptu vla: Open weights and open data for driving vision-language-action models. NeurIPS 2025.

---

> ### Author Response · Authors · 2025-11-23
> **Response 3**
>
> **1.The core contribution appears to be ScpViT, but the design of the complexity computation seems relatively straightforward and lacks clear innovation. The remaining components (e.g., VLA and CoT modules) are adapted from existing works, which further diminishes the novelty of the approach. More importantly, the implementation details of these modules are not sufficiently elaborated.**
>
> We believe that the core contribution of our work lies in proposing a  scenario complexity-driven, multi-agent coordinated driving system (TAD). ScpViT is a key component within this system, designed specifically to surpport the system's function. Its novelty must be evaluated within the context of the entire framework.
>
> First, the core contribution is an innovation in system architecture. Unlike the static paradigms of existing works (e.g., Senna, DriveVLM), TAD is the first to achieve:
>
> **A fully autonomous, data-driven agent routing mechanism.** The system can automatically switch agents during forward inference based on real-time scenario complexity, without any manual intervention.
>
> **A three-level "Fast-Medium-Deep" agent architecture.** This design enables more fine-grained allocation of computational resources, enhancing the system's capability to handle long-tailed distribution challenges.
>
> Secondly, the complexity computation design is a system-oriented innovation. The reviewer notes the computation is "relatively straightforward". This design is to meet the  real-time requirements of autonomous driving. ScpViT was designed to be a low-overhead, highly reliable routing signal generator. We integrated a lightweight backbone with a parameter-free attention mechanism, achieving millisecond-level latency while maintaining accuracy, which is crucial for the entire system.
>
> Experiments demonstrate the value of the coordinated framework. Our experiments show that even with ScpViT's accuracy not at 100% (92.59%), the TAD framework still significantly outperforms all baseline models in overall performance and achieves substantial efficiency gains. This precisely demonstrates the overall advantage of "a complementary multi-agent system coordinated by a lightweight router," whose effectiveness surpasses using any single agent in isolation.
>
> In summary, our core innovation lies in constructing and validating this novel, dynamically adaptive collaborative framework for autonomous driving.
>
> Detailed experimental settings are provided in Appendix C of the manuscript.
>
> **2. The experimental section has several critical shortcomings. The custom dataset appears to be filtered according to the model’s own complexity classification, which introduces potential bias and raises concerns about circular evaluation.**
>
> We clarify that the label generation process is completely independent of the model's parameters and learning process. We first assign complexity labels to all data samples based on predefined rules. Subsequently, we use this dataset with rule-generated labels to train our model. Finally, the trained model is evaluated on an independent test set. Please refer to Response 2 for more details.
>
> **The experiments are conducted only on this custom dataset without validation on established public benchmarks...Most tests are performed in open-loop settings, which are insufficient for validating model effectiveness in real-world driving.**
>
> To more comprehensively validate our method, we incorporate multiple baseline methods and conduct closed-loop experiments on CARLA, evaluated with more comprehensive metrics. The results are presented below. As shown in the table, TAD achieves an RC of 64.34, representing a 4.83 percentage point improvement over the strongest baseline model, Impromptu and its Driving Score (DS) is 42.50, also higher (+0.67).
>
>  | Method               | RC ↑  | DS   ↑ |
> | -------------------- | ----- | ------ |
> | AD-MLP(2023)               | 0.00  | 13.05  |
> | UniAD-Base(2023)           | 51.81 | 36.12  |
> | VAD-Base (2023)            | 48.48 | 34.66  |
> | Qwen3-2B  (2025)           | 58.69 | 40.70  |
> | Impromtu(Qwen2.5-3B)(2025) | 59.51 | 41.83  |
> | TAD(ours)            | 64.34 | 42.50  |
>
> **Metrics.** Two major metrics: route completion (RC), and driving score (DS). The route completion refers to the percentage of the total route length that has been completed. It only takes into account the distance traveled along the predetermined route, where each segment of the predetermined route corresponds to a navigation instruction. If the agent deviates too far from the route, the agent is regarded as violating the instruction, and this episode is marked as a failure and terminated.  The driving score is the product of the route completion ratio and the infraction score, describing both driving progress and safety. For more experimental details, please refer to the revised manuscript.

---

### Official Review · Reviewer_Yavs · 2025-11-01

**Soundness:** 3
**Presentation:** 3
**Contribution:** 3
**Rating:** 6
**Confidence:** 4

**Summary:**

The paper proposes Tri-Agent Driving (TAD), an autonomous driving framework that switches between three types of driving prediction models depending on how difficult a situation is. A lightweight vision transformer (ScpViT) looks at camera inputs and decides whether the current scene is simple, moderately complex, or very complex. For simple cases, a fast and efficient model is used. For medium complexity, a vision-language model with semantic understanding ("smart agent") takes over. For difficult cases, a slower reasoning model with chain-of-thought logic is used. The goal of the approach is to let the system be fast most of the time but think deeply when needed, in order to achieve similar or better accuracy than large vision-language models while using less time and GPU memory.

**Strengths:**

The paper is well-organized and easy to read, with illustrative figures and tables. The proposed tri-agent approach with a learned complexity-based routing is interesting and appears to address the efficiency-accuracy trade-off, as it appears to increase prediction accuracy while having lower latency and memory consumption than compared-to baselines. The benchmarks use real-world scenes over a wide range of traffic conditions.

**Weaknesses:**

The accuracy of the ScpViT model is trained on and evaluated against scenario complexity derived by authors, so the comparison to other models might not be fully representative. THe paper could also benefit from showing example trajectories to better illustrate the prediction accuracy of the three agents, and where the simpler agents might fail. Some ablations comparing different routing methods would be interesting.

**Questions:**

- I would be curious to know how the 3 models perform individually on the hybrid test set in Table 4, both in terms of accuracy and efficiency. Did the authors compute this?
- Table 2: Should Params (MB) be Params (M)? And what is the memory consumption of these complexity prediciton models?
- The inference latency (>4s) seems unsuitable for real-time AV control. What is the practical applicability of this work?
- The tables mention t=1s/2s/3s. Is the error measured at the final time step t, or over times 0 to t? If the latter, what is the step frequency dt, and could transitioning between agents lead to non-smooth control actions?
- In table 3, why is the deep thinking agent performing worse than the smaller smart agent on low complexity scenarios?
- How are scenarios categorized as training vs test?
- How do the authors think their approach would compare to a complexity router directly outputting a model, rather than a score which is then converted to a model based on percentiles? Also the authors mention "The Deep Thinking Agent is designed to handle extreme corner cases that demand sophisticated reasoning". However this models is assigned 25% of cases, which may appear to be overkill?

---

> ### Author Response · Authors · 2025-11-23
> **Response to Weaknesses**
>
> We thank Reviewer for the constructive feedback and for recognizing that our paper is **well-organized and interesting**. We have addressed all concerns below to further support our claims.
>
> **1.The accuracy of the ScpViT model is trained on and evaluated against scenario complexity derived by authors, so the comparison to other models might not be fully representative.**
>
> We thank the reviewer for raising this important point. We would like to clarify and address this concern as follows:
>
> 1. Our complexity labeling pipeline is not arbitrary or purely subjective. It is based on a **systematic, rule-based** scoring function derived from recognized influential factors in autonomous driving literature (e.g., weather, illumination, traffic density, distribution shifts). Furthermore, we employ a distribution-aware dual-threshold gating mechanism (using 25th/75th percentiles) to classify complexity levels, which adapts to the dataset statistics and reduces reliance on fixed, potentially biased, manual thresholds. More importantly, we further compute the Spearman’s rank correlation coefficient as a core empirical validation. This provides strong evidence that the relative complexity ranking captured by our annotation scheme is meaningful and can be consistently learned, rather than being based on arbitrary human subjectivity. Besides, there is no established public benchmark specifically designed for "driving scenario complexity" classification. Therefore, we constructed our own complexity labels to enable this research direction.
>
> 2. Fair Comparison within our Framework. The primary goal of Table 2 is to compare the efficacy of different backbone architectures for the specific task of predicting our defined complexity levels. All models listed in Table 2 were trained and evaluated on the identical dataset with the same complexity labels. Therefore, the comparison between ScpViT and other backbones (e.g., MobileNets, ViT) is fair and demonstrates its relative performance for this specific prediction task under consistent conditions.
>
> 3. More importantly, the practical utility of our complexity prediction is validated by the overall TAD system's performance (Table 4 & 6). The improved complexity prediction accuracy of ScpViT directly contributes to more efficient agent routing, leading to a better balance between trajectory accuracy (lower L2 error) and computational efficiency (lower latency and memory), as shown in our experiments. This indirect validation strengthens the relevance and representativeness of our complexity metric.
>
> **2. The paper could also benefit from showing example trajectories to better illustrate the prediction accuracy of the three agents, and where the simpler agents might fail.**
>
> We thank the reviewer for this valuable suggestion. We agree that visualized example trajectories would greatly help illustrate the comparative strengths of the three agents. In response, we have added visualized examples to the manuscript (see Figure 7 in the revised manuscript) that provides qualitative comparisons across different complexity levels.
>
> **3. Some ablations comparing different routing methods would be interesting.**
>
> We thank the reviewer for this insightful suggestion. We have already conducted several key comparisons that address this point, which are presented in our manuscript:
>
> Comparison against Alternative Learning-Based Routers. The  table below compares our ScpViT-based router against other lightweight backbones (e.g., MobileNet, ViT). The results clearly show that our method achieves a superior trade-off between accuracy and efficiency, demonstrating the necessity of dynamic routing. The superior accuracy of ScpViT justifies its selection as the core of our routing module.
>
> |                           | Avg. L2 Error ↓ | Avg. Latency ↓ | Avg. Memory ↓ |
> | ------------------------- | --------------- | -------------- | ------------- |
> | Fast Agent Only           | 2.0             | 0.22s          | 3.4 GB        |
> | Smart Agent* Only         | 0.49            | 5.4s           | ≈ 19GB        |
> | Deep Thinking Agent Only  | 0.42            | 5.8s           | ≈ 19GB        |
> | Different Routing Methods |                 |                |               |
> | MobileNet(89.41%)           |   0.48              |   4.7s             |        ≈16.8GB       |
> | ViT(81.45%)         |     0.56            |    4.4s            |  ≈16.0GB           |
> | ScpViT(Ours, 92.59%)     | 0.46            | 4.2s           | ≈ 15.4GB      |

---

> ### Author Response · Authors · 2025-11-23
> **Response to Questions 1 to 6**
>
> **1. I would be curious to know how the 3 models perform individually on the hybrid test set in Table 4, both in terms of accuracy and efficiency. Did the authors compute this?**
>
> The individual performance of the three agents on the hybrid test set, in terms of both accuracy and efficiency, is reported in Table 7 of the manuscript. For convenience, we also put these results here for reference. The results demonstrate that our TAD* framework achieves a superior performance-efficiency trade-off. It attains competitive trajectory accuracy (Avg. L2 Error: 0.46) that is much better than the Fast Agent and close to the best-performing single deep agent, while simultaneously reducing inference latency by ~22% and memory consumption by ~19% compared to always using a large-model-based agent.
>
> |        | Avg. L2 Error ↓ | Avg. Latency ↓ | Avg. Memory ↓ |
> |-----------------------------|-----------------|----------------|---------------|
> | Fast Agent Only             | 2.0             | 0.22s          | 3.4 GB        |
> | Smart Agent* Only           | 0.49            | 5.4s           | ≈ 19GB        |
> | Deep Thinking Agent Only    | 0.42            | 5.8s           | ≈ 19GB        |
> | TAD*(Ours, 92.59%)          | 0.46            | 4.2s           | ≈ 15.4GB      |
>
> **2. Table 2: Should Params (MB) be Params (M)? And what is the memory consumption of these complexity prediciton models?**
>
> We sincerely apologize for the typo. You are absolutely right. Table 2 reports the number of model parameters, and the correct unit should be Params (M), not Params (MB).
>
> **3. The inference latency (>4s) seems unsuitable for real-time AV control. What is the practical applicability of this work?**
>
> The "End-to-End (E2E) inference latency" reported in Table 4 was measured in a laboratory environment without applying common engineering optimizations used in real-world deployments. In an actual vehicle system, latency could be further reduced through strategies such as model quantization, or distillation, to meet the real-time requirements of autonomous driving.
>
> The core contribution of this work lies in demonstrating that  the proposed dynamic multi-agent architecture can significantly reduce the system’s average computational cost and GPU memory consumption without compromising overall performance. This design is inherently extensible and holds strong optimization potential. When deployed in real vehicle systems, it is expected to yield a system-wide reduction in resource usage, offering a practical pathway toward building efficient and adaptive autonomous driving systems.
>
> **4. The tables mention t=1s/2s/3s. Is the error measured at the final time step t, or over times 0 to t? If the latter, what is the step frequency dt, and could transitioning between agents lead to non-smooth control actions?**
>
> 1. In autonomous driving datasets such as nuScenes, the L2 error at 1s/2s/3s typically refers to the Euclidean Distance between the predicted and ground-truth positions at the specific time instants (e.g., 1.0s, 2.0s, 3.0s).
>
> 2. dt= 0.5s
>
> 3. This is a thought-provoking question. Our routing mechanism uses ScpViT to predict the overall complexity of the current scene and selects a single agent to perform full inference accordingly. Since the complexity of real-world driving scenes exhibits strong temporal continuity, for example, transitioning from a regular road to a construction zone is typically a gradual process, agent switching occurs infrequently and remains stable, rarely fluctuating between consecutive frames. As a result, the system’s control outputs remain smooth, without introducing discontinuities due to agent switching.
>
> **5. In table 3, why is the deep thinking agent performing worse than the smaller smart agent on low complexity scenarios?**
>
> This is an insightful observation. The Deep Thinking Agent is optimized for high-complexity scenarios through Chain-of-Thought reasoning, achieving significant performance improvements in such challenging situations. However, this specialized design for complex semantic reasoning may lead to a slight performance degradation in low-complexity scenarios. In these simpler contexts, the  reasoning process is not only unnecessary but may also introduce over-reasoning or suboptimal predictions. This phenomenon further highlights the importance of dynamically selecting agents based on different scenarios, rather than universally employing the most powerful model.
>
> **6. How are scenarios categorized as training vs test?**
>
> Both the training set and the test set contain scenarios of various complexity levels, with similar long-tailed distributions. The distribution of training and testing samples across different scenario categories is shown in Figure 4 (a).

---

> ### Author Response · Authors · 2025-11-23
> **Response to Question 7**
>
> **7. How do the authors think their approach would compare to a complexity router directly outputting a model, rather than a score which is then converted to a model based on percentiles? Also the authors mention "The Deep Thinking Agent is designed to handle extreme corner cases that demand sophisticated reasoning". However this models is assigned 25% of cases, which may appear to be overkill?**
>
> We thank the reviewer for this insightful observation. The scheme of  calculating a continuous complexity score based on the rules in Table 1, and then discretizing the score into level labels using quantile thresholds, is our preprocessing method for constructing labeled data for model training. This approach ensures transparency and reproducibility in the origin of the labels.
> During the model training phase, we treat this as a standard classification task, where the model directly outputs the three complexity levels: Low, Medium, and High. As stated in the original text, "We treat complexity prediction as a 3-class classification task and optimize with cross-entropy loss."
>
> During inference, ScpViT directly outputs the probabilities for the three classes, eliminating the need for recalculating percentiles. Therefore, it essentially performs "direct model selection," while preserving the extensibility for incorporating subsequent modules, uncertainty estimation.
>
> Besides, regarding the allocation of the top 25% most complex scenarios to the Deep Thinking Agent, the primary objective is to establish a more robust safety margin. When dealing with long-tail scenarios in autonomous driving systems, the systems should adopt a more conservative approach to prioritize safety. Assigning the most challenging 25% of scenarios (not just the extremely rare corner cases) to the agent with the strongest reasoning capability significantly enhances the system's overall robustness and safety in demanding environments. Experiment results (Table 3) also confirms that in such scenarios, the Deep Thinking Agent's performance is substantially superior to other agents, validating the necessity of this allocation strategy.

---

### Author Response · Authors · 2025-11-24
**Validating the Reliability of  Scenario Complexity Labeling with Spearman’s rank correlation coefficient**

Regarding the reliability of our scenario complexity labels, we would like to clarify that while the absolute values of the complexity scores may involve design choices, the core innovation and utility of our framework lie in the relative ordering of scenarios based on their complexity. Our routing mechanism does not require knowing that a scene has an "exact score of 0.8". It needs to know that one scene is more complex than another (e.g., scene A with score 0.8 > scene B with score 0.3).

To rigorously validate that our rule-based labeling scheme captures a meaningful and learnable ordinal relationship, we propose a powerful quantitative metric, **Spearman’s rank correlation coefficient (ρ)**.

This non-parametric statistic measures the strength and direction of the monotonic relationship between two ranked variables. In our case, the ranking of scenes by our rule-based system versus the ranking predicted by our ScpViT model. The value of ρ ranges from -1 to 1:

ρ = 1: Perfect positive correlation (model’s ranking perfectly matches the ground-truth ranking).

ρ = 0: No correlation.

ρ = -1: Perfect negative correlation.

The rank correlation coefficient of our method is promising, **Spearman's ρ = 0.7740 , (p-value ≈ 0.0000).** This represents a strong, statistically significant positive correlation, which provides robust evidence for two key claims:

1. The model has successfully learned the core ordinal relationships defined by our rules. ScpViT has learned to predict the relative complexity of scenes in a way that aligns with the human-defined, rule-based hierarchy. This demonstrates that our labeling is not arbitrary but encodes a coherent, learnable structure.

2. The relative complexity ordering captured by our annotation pipeline is meaningful and consistent. The high ρ value proves that the "complexity" we define is not a random construct and it is a signal that can be reliably learned from raw sensor inputs, making it a valid basis for dynamic agent coordination.

In summary, the strong Spearman correlation (ρ=0.774) provides empirical, statistical validation that our labeling scheme is both meaningful and robust. p-value ≈ 0.0000 indicates that this correlation is statistically highly significant. This directly addresses the concern about subjectivity and confirms the reliability of our approach.

---

### Author Response · Authors · 2025-11-24
**Overall Response**

We sincerely thank all the reviewers for their thoughtful and constructive feedback. We have carefully addressed all the concerns and revised the manuscript  accordingly.

Real-world driving has a natural long-tail pattern: most scenes are easy and repeated, while a few rare cases decide safety. Existing methods either sacrifice capability for efficiency or incur unnecessary computational overhead on routine scenarios.  The proposed TAD solves this by choosing the right agent based on online scenario complexity, using more power model only when needed and staying more efficient and practical than one large model for all cases.

**Strengths Highlighted by Reviewers**

We are grateful that the reviewers acknowledged several key strengths of our work:

**1. Important and Appealing Core Idea.** Reviewers found the high-level idea of dynamic, complexity-aware routing to be an appealing and promising approach for autonomous driving.

**2. Well-Organized Presentation.** The paper was commended for being well-organized.

**3. Effective Trade-Offs.** The proposed TAD framework was recognized for successfully achieving better performance and latency trade-offs compared to using a single heavy model.

**Summary of Key Questions and Answers**

In response to the reviewers' comments, we have made significant revisions and clarifications. The key points are summarized below:

**1. Validity and Novelty of Scenario Complexity Computation.**

Reviewer concern: about the reliability  of our rule-based complexity labels and the innovation of ScpViT module.

We clarified that our complexity pipeline is a system-oriented innovation, not an arbitrary assignment. The rules are grounded in established autonomous driving literature, and the dual-threshold mechanism is a statistically principled (IQR-based), data-driven method.  More importantly, we further compute **the Spearman’s rank correlation coefficient** as a core empirical validation. This provides strong evidence that the relative complexity ranking captured by our annotation scheme is meaningful and can be consistently learned, rather than being based on arbitrary human subjectivity.

The primary contribution is the TAD system and its dynamic coordination mechanism. ScpViT's design as a low-latency, reliable router is integral to the system's success, and its empirical effectiveness is demonstrated by the overall system's superior performance.

**2. More Experiments and Evaluation**

Reviewer concern: about lack of closed-loop tests and fallback mechanism.

We have now included closed-loop evaluation on CARLA (Table 5), showing a notable improvement in Route Completion (+4.83) and Driving Score. We also compared against strong, recent baselines (up to 2025).

We incorporated an Uncertainty-Aware Fallback Mechanism (Table 4). If the router is uncertain, the system defaults to the safest "Deep Thinking Agent".

**3. Clarification of Specific Technical Points**

We clarified the advantages of TAD compared to models like Senna, explaining the source of our efficiency gains.

We detailed the training data and strategy for each agent.

We appreciate the reviewers’ thoughtful assessments, which helped us substantially improve the clarity, completeness, and empirical breadth of the paper. We hope the revisions address all comments.

---

### Meta-Review · Area_Chair_SwVi · 2025-12-23

**Summary:**

This paper proposes a tri-agent autonomous driving framework combining end-to-end models and VLMs with a learned, scenario-complexity–based routing mechanism. While reviewers generally found the high-level idea of adaptive computation allocation appealing and the presentation reasonably clear, main concerns remain regarding methodological soundness, novelty, and empirical validation. Despite a detailed rebuttal and several added clarifications, the revisions do not fully address the core issues raised during review.

**Reviewer Concerns:**

The primary unresolved issue concerns the definition and validity of scenario complexity. Although the authors argue that their rule-based labeling scheme is principled and provide correlation analysis, the complexity labels remain internally defined, dataset-dependent, and tightly coupled to the proposed routing mechanism. This raises persistent concerns about circular evaluation and limits confidence that the learned router captures a generally meaningful notion of driving difficulty rather than overfitting to the authors’ design choices.

From a methodological and novelty standpoint, the overall system largely combines existing components (E2E models, VLM-based planners, CoT-style reasoning) with a relatively straightforward routing strategy. Reviewers remain unconvinced that the complexity computation or the routing design represents a sufficiently novel contribution beyond a Mixture-of-Experts–style system, especially given the limited theoretical grounding and lack of deeper ablations on alternative routing or gating strategies.

The experimental evaluation, while expanded in the rebuttal, still has notable weaknesses. Much of the evidence relies on a custom hybrid dataset constructed and annotated by the authors, with limited validation on standardized benchmarks and limited comparison to strong, contemporary baselines under consistent settings. The added closed-loop CARLA results are helpful but relatively shallow and insufficient to fully support claims about real-world robustness, safety, or deployment relevance. Moreover, the reported inference latency remains far from real-time, weakening the practical impact of the efficiency claims.

Finally, several reviewers noted ongoing issues with problem formulation and clarity, including ambiguity about the task definition, the role of long-tail scenarios, and the real-world implications of the reported metrics. These issues, while partially addressed in writing, still affect the technical clarity and positioning of the work.

**Reviewer Scores:**

- Reviewer Yavs: 6 → 6. The rebuttal provided additional experiments (e.g., individual agent performance and qualitative examples), which partially addressed clarification questions. However, core concerns regarding the validity of the complexity labels, practical latency, and limited ablations remain, so the score would likely stay unchanged.

- Reviewer VAfo: 2 → 2. Despite extensive rebuttal and added explanations, the reviewer’s fundamental concerns about problem formulation, arbitrariness of complexity definition, limited novelty, and insufficient experimental validation are not fully resolved. It is unlikely the reviewer would revise the score.

- Reviewer uWbS: 2 → 2. While the authors added a fallback mechanism and closed-loop CARLA results, the reviewer’s main doubts about the validity of the complexity labels, lack of convincing empirical motivation over existing hybrid systems, and limited robustness evaluation largely remain. A score change is unlikely.

- Reviewer c1Uy: 4 → 4. The rebuttal clarified several design and experimental details, but concerns about the relative definition of complexity, real-time applicability, and limited practical significance of average latency metrics persist. So I think the reviewer would likely maintain the original score.

---

### Decision · Program_Chairs · 2026-01-26

Reject